# GENERATION, RECONSTRUCTION, REPRESENTATION ALL-IN-ONE: A JOINT AUTOENCODING DIFFUSION MODEL

## ABSTRACT

The vast applications of deep generative models are founded on the premise of three fundamental capabilities: *generating* new instances (e.g., image/text synthesis and molecule design), *reconstructing* inputs (e.g., data editing and restoration), and learning latent *representations* (e.g., structure discovery and downstream classification). Existing model families, including Variational Autoencoders (VAEs), Generative Adversarial Networks (GANs), autoregressive models, and diffusion models, generally excel in specific capabilities but fall short in others. We introduce Joint Autoencoding Diffusion (JEDI), a new generative framework that unifies all three core capabilities, offering versatile applications and strong performance in a single model. Specifically, JEDI generalizes the noising/denoising transformations (based on simple Gaussian noise) in diffusion process by introducing *parameterized* encoder/decoder transformations between raw data and compact representations. Crucially, the encoder/decoder parameters are learned *jointly* with all other diffusion model parameters under the standard probabilistic diffusion formalism. This results in a model that not only inherits the strong generation abilities of diffusion models but also enables compact data representation and faithful reconstruction. Additionally, by choosing appropriate encoder/decoder, JEDI can naturally accommodate discrete data (such as text and protein sequences) which have been difficult for diffusion models. Extensive experiments across different data modalities, including images, text, and proteins, demonstrate JEDI's general applicability to diverse tasks and strong improvement over existing specialized deep generative models.

## 1 INTRODUCTION

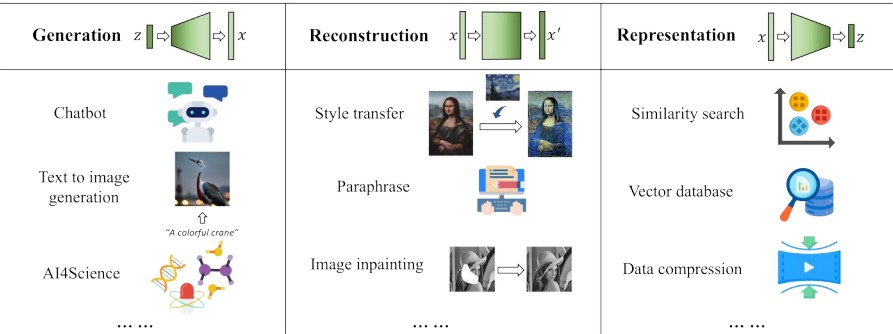

Figure 1: An illustration of three fundamental abilities of deep generative models.

Generative models are central to deep learning, offering the ability to simulate, understand, and manipulate complex data distributions. Their widespread applications hinge on three fundamental capabilities: 1) *Generating* new instances aligned with a learned distribution; 2) *Reconstructing* the inputs, which involves the extraction and utilization of salient features to reproduce input data. 3) Learning latent *representation* that enables structure discovery and downstream classification. Fig. 1 underscores the importance of three capabilities by highlighting their respective applications.

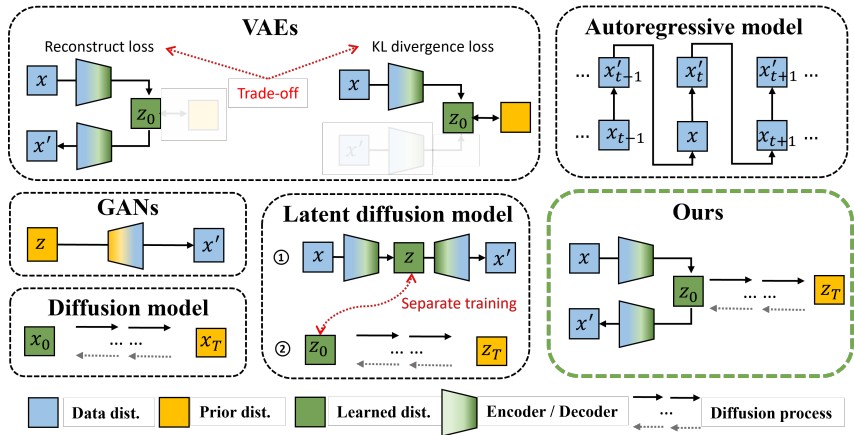

Figure 2: Conceptual illustrations of existing generative models and our proposed model.

Deep generative models, like Variational Autoencoders (VAEs, Kingma & Welling (2014)), Generative Adversarial Networks (GANs, Goodfellow et al. (2014)), autoregressive models (Van Den Oord et al., 2016), and Diffusion models (Sohl-Dickstein et al., 2015; Ho et al., 2020), each exhibits specific strengths in various applications. However, a comprehensive model that seamlessly integrates all three fundamental capabilities remains elusive.

VAEs are known for their trade-off between reconstruction and generation, a balance governed by the reconstruction loss and regularization loss (Bowman et al., 2016; Chen et al., 2017). While variants such as the $\beta$-VAE (Higgins et al., 2016) aim to offer a more controllable trade-off, they continue to face challenges in simultaneously achieving high-quality generation and precise reconstruction. GANs, while renowned for their impressive generation capabilities, inherently lack an explicit representation of the latent space (Radford et al., 2015; Donahue et al., 2016; Chen et al., 2016). Although subsequent research (Zhou et al., 2019; Xia et al., 2023) has incorporated encoders into GANs for reconstruction, as seen in BiGANs (Donahue et al., 2016) and CycleGANs (Zhu et al., 2017), achieving a harmonious balance between generation, reconstruction, and meaningful latent representation is still challenging. Autoregressive models excel in generating content, particularly in the text domain (Vaswani et al., 2017; Radford et al., 2019). However, they often produce outputs with limited diversity and do not inherently offer a semantically rich latent space for representation. Diffusion models are adept at generation but frequently fall short in providing a semantically rich latent space and effective reconstruction. While DDIM (Song et al., 2021) enhances the reconstruction capabilities of diffusion models, it still fail to encapsulate high-level semantics essential for meaningful representations (Preechakul et al., 2022; Xiang et al., 2023). Latent diffusion model (Rombach et al., 2021) employs an autoencoder to represent data in a low-dimensional space for compression and enhanced efficiency, yet it still does not incorporate meaningful semantics.

Addressing these challenges, our primary contribution is the introduction of a unified generative model: Joint Autoencoding Diffusion (JEDI, §3), tailored to tackle all three core functionalities. Specifically, we generalize the noising/denoising transformations in standard diffusion process by introducing *parameterized* encoder/decoder transformations between raw data and compact representations. In the forward process, the encoder introduces a specific "noise" and concurrently maps data into a compact latent space. In contrast, the decoder operates in the reverse, aiming to predict and counteract the noise introduced by the encoder. This generalization naturally complements the established diffusion model framework. Building upon these generalizations, we formulate the training objective directly from the data likelihood. Under the standard probabilistic diffusion formalism, the encoder/decoder parameters are learned *jointly* with all other diffusion model parameters. This results in a model that not only inherits the strong generation abilities of diffusion models but also enables compact data representation and faithful reconstruction. Given the inherent fixed latent dimensionality of conventional diffusion models, handling variable-length sequential data remains a challenge. However, by selecting suitable encoder/decoder configurations, our model can seamlessly processes discrete data types, like text and protein sequences. A conceptual comparison between our model and existing generative models can be seen in Fig. 2.

Our comprehensive experiments (§4), spanning varied data modalities including images, text, and protein sequences, demonstrate our model's general applicability to diverse tasks and strong improvement over existing specialized deep generative models.

## 2 BACKGROUND

### 2.1 DIFFUSION MODELS

Diffusion models (Sohl-Dickstein et al., 2015) are probabilistic models designed to approximate a target data distribution $q(\mathbf{x}_0)$ with a model distribution $p_\theta(\mathbf{x}_0)$ by gradually denoising a normally distributed variable. The predominant formulation is the Denoising Diffusion Probabilistic Models (DDPMs, Ho et al. (2020)), which involve two phases: the forward process and the reverse process.

**Forward Process**   The forward process aims to transform any data distribution into a simple standard Gaussian distribution. This process, typically hand-designed, is formulated as a Markov chain that sequentially introduces Gaussian noise into the data, as defined by:

$$q(\mathbf{x}_{1:T}|\mathbf{x}_0) := \prod_{t=1}^{T} q(\mathbf{x}_t|\mathbf{x}_{t-1}); \quad q(\mathbf{x}_t|\mathbf{x}_{t-1}) := \mathcal{N}(\mathbf{x}_t; \sqrt{1-\beta_t}\mathbf{x}_{t-1}, \beta_t\mathbf{I}), \tag{1}$$

where $T$ is the number of diffusion steps, $\beta_1, \ldots, \beta_T$ represent the pre-defined variance schedule, $\mathbf{x}_1, \ldots, \mathbf{x}_T$ are latent vectors of the same dimensionality as the data $\mathbf{x}_0 \sim q(\mathbf{x}_0)$, and $\mathbf{x}_T$ admits the standard Gaussian distribution when $T \to \infty$.

**Reverse Process**   The reverse process involves learning $p_\theta(\mathbf{x}_{t-1}|\mathbf{x}_t)$ to approximate the intractable $q(\mathbf{x}_{t-1}|\mathbf{x}_t)$. When $\beta_t$ is small enough, $q(\mathbf{x}_{t-1}|\mathbf{x}_t)$ adopts a Gaussian form, ensuring the expressiveness of the reverse process. Initialized at $p(\mathbf{x}_T) = \mathcal{N}(\mathbf{x}_T; \mathbf{0}, \mathbf{I})$, the reverse process is formulated as a Markov chain featuring learnable Gaussian transitions:

$$p_\theta(\mathbf{x}_{0:T}) := p_\theta(\mathbf{x}_T) \prod_{t=1}^{T} p_\theta(\mathbf{x}_{t-1}|\mathbf{x}_t), \quad p_\theta(\mathbf{x}_{t-1}|\mathbf{x}_t) := \mathcal{N}(\mathbf{x}_{t-1}; \boldsymbol{\mu}_\theta(\mathbf{x}_t, t), \boldsymbol{\Sigma}_\theta(\mathbf{x}_t, t)). \tag{2}$$

**Training Objective**   During training, the parameters $\theta$ are optimized via the variational lower bound, which provides a bound on the marginal likelihood:

$$\mathbb{E}_q[-\log p_\theta(\mathbf{x}_0)] \le \mathbb{E}_q\left[-\log\frac{p_\theta(\mathbf{x}_{0:T})}{q(\mathbf{x}_{1:T}|\mathbf{x}_0)}\right] = \mathbb{E}_q\left[-\log p(\mathbf{x}_T) - \sum_{t=1}^{T}\log\frac{p_\theta(\mathbf{x}_{t-1}|\mathbf{x})}{q(\mathbf{x}_t|\mathbf{x}_{t-1})}\right] =: \mathcal{L}(\theta). \tag{3}$$

Although $q(\mathbf{x}_t|\mathbf{x}_{t-1})$ is intractable on its own, it becomes tractable when conditioned on $\mathbf{x}_0$:

$$q(\mathbf{x}_{t-1}|\mathbf{x}_t, \mathbf{x}_0) = \mathcal{N}(\mathbf{x}_{t-1}; \tilde{\boldsymbol{\mu}}_t(\mathbf{x}_t, \mathbf{x}_0), \tilde{\beta}_t\mathbf{I})), \tag{4}$$

$$\text{where } \tilde{\boldsymbol{\mu}}_t(\mathbf{x}_t, \mathbf{x}_0) := \frac{\sqrt{\bar{\alpha}_{t-1}}\beta_t}{1-\bar{\alpha}_t}\mathbf{x}_0 + \frac{\sqrt{\alpha_t}(1-\bar{\alpha}_{t-1})}{1-\bar{\alpha}_t}\mathbf{x}_t \text{ and } \tilde{\beta}_t := \frac{1-\bar{\alpha}_{t-1}}{1-\bar{\alpha}_t}\beta_t. \tag{5}$$

Here $\alpha_t := 1 - \beta_t$ and $\bar{\alpha}_t := \prod_{s=1}^{t}\alpha_s$. Then $\mathcal{L}(\theta)$ can be rendered tractable by rewriting it as:

$$\mathbb{E}_q[\underbrace{\text{KL}\left(q(\mathbf{x}_T|\mathbf{x}_0)||p(\mathbf{x}_T)\right)}_{\mathcal{L}_T} + \sum_{t=2}^{T}\underbrace{\text{KL}\left(q(\mathbf{x}_{t-1}|\mathbf{x}_t, \mathbf{x}_0)||p_\theta(\mathbf{x}_{t-1}|\mathbf{x}_t)\right)}_{\mathcal{L}_{t-1}} - \underbrace{\log p_\theta(\mathbf{x}_0|\mathbf{x}_1)}_{\mathcal{L}_0}]. \tag{6}$$

$\mathcal{L}_T$ is ignored during optimization since it contains no learnable parameters. Both $\mathcal{L}_{t-1}$ and $\mathcal{L}_0$ are reduced to

$$\mathbb{E}_{\mathbf{x}_0 \sim q(\mathbf{x}_0), \boldsymbol{\epsilon} \sim \mathcal{N}(\mathbf{0}, \mathbf{I})}\left[\gamma_t\|\boldsymbol{\epsilon} - \boldsymbol{\epsilon}_\theta(\sqrt{\bar{\alpha}_t}\mathbf{x}_0 + \sqrt{1-\bar{\alpha}_t}\boldsymbol{\epsilon}, t)\|^2\right] + C \tag{7}$$

by reparemetrizing $\boldsymbol{\mu}_\theta(\mathbf{x}_t, t)$ as $\frac{1}{\sqrt{\bar{\alpha}_t}}(\mathbf{x}_t - \sqrt{1-\bar{\alpha}_t}\boldsymbol{\epsilon}_\theta(\mathbf{x}_t, t))$ and approximating $\mathcal{L}_0$ on discrete log likelihood, where $\gamma_t = \frac{\beta_t^2}{2\sigma_t^2\alpha_t(1-\bar{\alpha}_t)}$ and $C$ is a constant. During training, the loss is further simplified by setting $\gamma_t$ as 1 (Ho et al., 2020).

### 2.2 VARIATIONAL AUTOENCODERS (VAEs)

VAEs are a type of generative model that employ Bayesian inference techniques to learn a probabilistic mapping between the data space and a latent space (Kingma & Welling, 2014; Rezende et al., 2014). The key idea is to maximize the evidence lower bound (ELBO) with respect to both an encoder, which maps data to the latent space, and a decoder, which reconstructs data from the latent space:

$$\mathcal{L}(\lambda, \phi; \mathbf{x}) = -\mathbb{E}_{q_\lambda(\mathbf{z}|\mathbf{x})}[\log p_\phi(\mathbf{x}|\mathbf{z})] + \text{KL}(q_\lambda(\mathbf{z}|\mathbf{x})||p(\mathbf{z})), \tag{8}$$

where the two terms refer to the reconstruction loss and regularization loss, respectively.

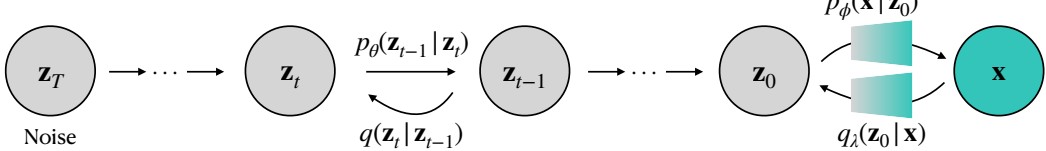

Figure 3: Overview of the diffusion process. The terms $q_\lambda(\mathbf{z}_0|\mathbf{x})$, $p_\phi(\mathbf{x}|\mathbf{z}_0)$, and $p_\theta(\mathbf{z}_{t-1}|\mathbf{z}_t)$ correspond to the diffusion-based encoder, decoder and latent state transition model, respectively.

## 3 JOINT AUTOENCODING DIFFUSION (JEDI)

To address aforementioned challenges, we present JEDI, a comprehensive generative model that unifies generation, reconstruction, and representation in one single model. Specifically, JEDI seamlessly integrates variational auto-encoding step into a generalized diffusion process by introducing parameterized encoder/decoder transformations between raw data and latent representations, as illustrated in Fig. 3 and §3.1. Aligning with the insights of Bansal et al. (2022), JEDI interprets encoding as noise addition and decoding as denoising, achieving a reversible design. Under the standard variational inference framework, the encoder/decoder parameters are learned jointly with all other diffusion model parameters (§3.2). Crucially, JEDI not only inherits the strong generation abilities of diffusion models but also enables compact data representation and faithful reconstruction through VAEs (§3.3) by modeling the diffusion process as a learnable prior for VAE (§3.4).

### 3.1 DIFFUSION PROCESS

**Forward Process** We incorporate an encoder $\mathcal{E}_\lambda(\cdot)$ to map data $\mathbf{x}$ into a low-dimensional latent space $\mathcal{Z}_0$, which is integrated as an additional forward step to the established Markov chains on $\mathcal{Z}_t$s, for $t \in 1, ..., T$. The newly defined forward process can be expressed mathematically as:

$$q_\lambda(\mathbf{z}_{0:T}|\mathbf{x}) := q_\lambda(\mathbf{z}_0|\mathbf{x}) \prod_{t=1}^T q(\mathbf{z}_t|\mathbf{z}_{t-1}) \tag{9}$$

$$q_\lambda(\mathbf{z}_0|\mathbf{x}) := \mathcal{N}(\mathbf{z}_0; \mathcal{E}_\lambda(\mathbf{x}), \beta_0\mathbf{I}), \quad q(\mathbf{z}_t|\mathbf{z}_{t-1}) := \mathcal{N}(\mathbf{z}_t; \sqrt{1-\beta_t}\mathbf{z}_{t-1}, \beta_t\mathbf{I}), \tag{10}$$

where $\mathbf{z}_t \in \mathcal{Z}_t, \forall t \in 0, ..., T$, and $\mathbf{z}_T$ is expected to admit the standard Gaussian distribution.

**Reverse Process** Commencing from an initial distribution $p(\mathbf{z}_T) = \mathcal{N}(\mathbf{z}_T; \mathbf{0}, \mathbf{I})$, the encompassing probabilistic formulation, encapsulating both the observed data and latent variables across the entire sequence, is thus:

$$p_{\phi,\theta}(\mathbf{x}, \mathbf{z}_{0:T}) = p_\phi(\mathbf{x}|\mathbf{z}_0)p(\mathbf{z}_T) \prod_{t=1}^T p_\theta(\mathbf{z}_{t-1}|\mathbf{z}_t) \tag{11}$$

$$p_\phi(\mathbf{x}|\mathbf{z}_0) = \mathcal{N}(\mathbf{x}; \mathcal{D}_\phi(\mathbf{z}_0), \mathbf{0}), \quad p_\theta(\mathbf{z}_{t-1}|\mathbf{z}_t) = \mathcal{N}(\mathbf{z}_t; \boldsymbol{\mu}_\theta(\mathbf{z}_t, t), \sigma_t\mathbf{I}), \tag{12}$$

where $p_\phi(\mathbf{x}|\mathbf{z}_0)$ designates the decoder $\mathcal{D}_\phi(\cdot)$ and $p_\theta(\mathbf{z}_{t-1}|\mathbf{z}_t)$ signifies the transition probabilities in the latent space, dictated by $\theta$.

### 3.2 TRAINING OBJECTIVE

Compared to DDPM, we extend the likelihood function to the probability determined by both the diffusion parameters $\theta$ and the decoder parameters $\phi$, corresponding to model the observed data through diffusion process followed by decoding process (details in §A.1):

$$\mathbb{E}_q[-\log p_{\phi,\theta}(\mathbf{x})] \leq \mathbb{E}_q\left[-\log\frac{p_{\phi,\theta}(\mathbf{x}, \mathbf{z}_{0:T})}{q_\lambda(\mathbf{z}_{0:T}|\mathbf{x})}\right] =: \mathcal{L}(\lambda, \phi, \theta) \tag{13}$$

This objective is aimed at the concurrent optimization of $\lambda$ for the encoder, $\phi$ for the decoder, as well as $\theta$ governing the latent state transitions in the diffusion model (details in §A.2):

$$\mathcal{L}(\lambda, \phi, \theta) = \mathbb{E}_q\left[\underbrace{\text{KL}\left(q(\mathbf{z}_T|\mathbf{z}_0)||p(\mathbf{z}_T)\right)}_{\mathcal{L}_T} + \sum_{t=2}^T \underbrace{\text{KL}\left(q(\mathbf{z}_{t-1}|\mathbf{z}_t, \mathbf{z}_0)||p_\theta(\mathbf{z}_{t-1}|\mathbf{z}_t)\right)}_{\mathcal{L}_{t-1}}\right.$$
$$\left. + \underbrace{\text{KL}\left(q_\lambda(\mathbf{z}_0|\mathbf{x})||p_\theta(\mathbf{z}_0|\mathbf{z}_1)\right)}_{\mathcal{L}_{\text{align}}} - \underbrace{\log p_\phi(\mathbf{x}|\mathbf{z}_0)}_{\mathcal{L}_{\text{rec}}}\right]. \tag{14}$$

Both $\mathcal{L}_T$ and $\mathcal{L}_{t-1}$ align with the terms in Eq. (6). This alignment can be achieved by simply substituting $\mathbf{x}$ with $\mathbf{z}$ (Details in §A.3). The distinction between our loss and that of DDPM (Eq. (6)) lies in the last two terms: $\mathcal{L}_{\text{align}}$ and $\mathcal{L}_{\text{rec}}$.

$\mathcal{L}_{\textbf{align}}$   This loss serves to align the latent representations derived from both the encoder and the diffusion model, ensuring consistent latent spaces across the framework. Specifically, the KL divergence between Gaussians $q_\lambda(\mathbf{z}_0|\mathbf{x})$ and $p_\theta(\mathbf{z}_0|\mathbf{z}_1)$, as delineated in Eq. (10) and Eq. (12), can be expressed as:

$$\mathcal{L}_{\text{align}} = \mathbb{E}_q \left[ \frac{1}{2\sigma_1^2} \|\mathcal{E}_\lambda(\mathbf{x}) - \boldsymbol{\mu}_\theta(\mathbf{z}_1, 1)\|^2 \right] + C = \mathbb{E}_q \left[ \gamma_1 \cdot \rho \|\mathcal{E}_\lambda(\mathbf{x}) - \boldsymbol{\mu}_\theta(\mathbf{z}_1, 1)\|^2 \right] + C, \quad (15)$$

where $C$ is a constant, and $\rho := \frac{\alpha_1(1-\bar{\alpha}_1)}{\beta_1^2}$ is introduced to align with Eq. (7).

$\mathcal{L}_{\textbf{rec}}$   This terms aligns with the reconstruction loss of VAE in Eq. (8). It is formulated as MSE loss for continuous data like images and cross-entropy for discrete data like texts and proteins.

The final objective $\mathcal{L}^{\text{final}}$ can be reformulated as (Details in §A.4):

$$\mathbb{E}_{q,\boldsymbol{\epsilon}} \left[ \sum_{t=2}^{T} \gamma_t \underbrace{\left( w \|\boldsymbol{\epsilon} - \boldsymbol{\epsilon}_\theta(\mathbf{z}_t, t)\|^2 + \frac{1}{\gamma_t T} \mathcal{L}_{\text{rec}} \right)}_{\mathcal{L}_t^{\text{final}}} + \gamma_1 \underbrace{\left( w \left( \rho \|\mathcal{E}_\lambda(\mathbf{x}) - \boldsymbol{\mu}_\theta(\mathbf{z}_1, 1)\|^2 \right) + \frac{1}{\gamma_1 T} \mathcal{L}_{\text{rec}} \right)}_{\mathcal{L}_1^{\text{final}}} \right], \quad (16)$$

where we set the weights of $\mathcal{L}_t^{\text{final}}$ to $1, \forall t \in \{1, ..., T\}$, following Ho et al. (2020) for further simplification, and $w$ is a hyperparameter introduced to balance the diffusion/align loss with the construction loss. The complete training approach is detailed in Algorithm 1 within Appendix B.

## 3.3 Core Functionalities: Reconstruction, Generation, and Representation

Our proposed diffusion model is designed to excel in three pivotal tasks: *generating* diverse new samples, accurately *reconstructing* input data, and *representing* data in a rich latent space. For *generation*, starting with a random sample $\mathbf{z}_T$ from $p(\mathbf{z}_T)$, the model undergoes a reverse diffusion and leverages the decoder to produce a fresh sample $\hat{\mathbf{x}}$, expressed as:

$$\mathbf{z}_T \sim p(\mathbf{z}_T), \qquad \mathbf{z}_0 \leftarrow \text{Reverse Diffusion}(\mathbf{z}_T), \qquad \text{and} \quad \hat{\mathbf{x}} = \mathcal{D}_\phi(\mathbf{z}_0). \quad (17)$$

This stochastic methodology ensures diverse sample production. For *reconstruction*, an input $\mathbf{x}$ is encoded to a latent representation and decoded back, producing $\hat{\mathbf{x}} = \mathcal{D}_\phi(\mathcal{E}_\lambda(\mathbf{x}))$. The fidelity of this process can be assessed via metrics like MSE, which is crucial for data interpolation and manipulation. For *representation*, inputs are transformed into a semantically profound latent space $\mathcal{Z}$, generating beneficial embeddings for a range of tasks. The efficacy of this representation hinges on its ability to retain essential features and its versatility in subsequent operations.

## 3.4 A Learnable Prior Perspective

The conventional VAE (§2.2) employs a standard Gaussian as its prior. However, achieving this prior proves challenging (Doersch, 2016; Bowman et al., 2016; Kingma et al., 2017). Despite the KL term in VAEs striving to narrow the distance between the posterior and the prior, a significant gap often persists. This gap can compromise the quality of the generated outcomes.

A promising approach to address this discrepancy is to introduce a learnable prior in place of the standard Gaussian (Dilokthanakul et al., 2016; Chen et al., 2017; Tomczak & Welling, 2018; Razavi et al., 2019; Wehenkel & Louppe, 2021; Vahdat et al., 2021). From this perspective, our methodology can be interpreted as an extension of the VAE, augmented with a learnable prior. Diverging from Eq. (8), our objective can be recast using Eq. (13) as (See §A.5 for details):

$$\mathcal{L}(\lambda, \phi, \theta; \mathbf{x}) = -\mathbb{E}_{q_\lambda(\mathbf{z}_0|\mathbf{x})}[\log p_\phi(\mathbf{x}|\mathbf{z}_0)] + \text{KL}(q_\lambda(\mathbf{z}_0|\mathbf{x})||p_\theta(\mathbf{z}_0)) \quad (18)$$

In our proposed formulation, the prior is distinctively parameterized by $p_\theta(\mathbf{z}_0)$, which harnesses the capabilities of a diffusion model in the latent space. This parameterization not only endows the VAE model with enhanced flexibility but also amplifies the representation capacity of the latent variables. Reciprocally, this configuration facilitates the diffusion model's adaptability to variable-length inputs, accommodating diverse data types like texts and protein sequences.

# 4 EXPERIMENTS

We evaluate JEDI's performance across all three fundamental capabilities using diverse modalities of data: image (§4.1), text (§4.2), and protein sequences (§4.3). For each modality, we consistently employ a simple MLP with skip connections as the diffusion model in the latent space, as suggested by Preechakul et al. (2022). By choosing appropriate encoder/decoder, JEDI seamlessly adapts to these diverse modalities. Our experiments across these modalities highlight JEDI's excellence in the three fundamental capabilities.

## 4.1 IMAGE

**Setup**  We adopt UNet (Long et al., 2015) as the encoder and the diffusion-based model in DiffAE (Preechakul et al., 2022)) as the decoder. Following DiffAE, we train our model and then evaluate the reconstruction as well as generation ability on FFHQ (Karras et al., 2019) and CelebA (Karras et al., 2018). Then we use the CelebA-HQ dataset to test the representation ability. In addition to DiffAE, we benchmarked our model against the Latent Diffusion Model (LDM, (Rombach et al., 2021)) and DDIM (Song et al., 2021) for comparative analysis. We use FID and reconstruction-FID (rFID) to evluate the generation quality and reconstruction quality, respecitively. To synthetically evaluate JEDI, besides the individual evaluations of reconstruction and generation (§4.1.1), we mainly perform image *interpolation* (§4.1.2) and *manipulation* (§4.1.3) tasks, which can reflect the comprehensive ability. A detailed image experimental setup can be found in §C.1.1.

### 4.1.1 IMAGE GENERATION AND RECONSTRUCTION

We evaluate generation and reconstruction performance across varying inference steps $T$. A comprehensive summary of our results is presented in Table 1. Notably, JEDI outperforms the baselines, showing improvements in both generation and reconstruction, which underscores the effectiveness of JEDI.

| Dataset | Model | FID | | | rFID | | |
|---|---|---|---|---|---|---|---|
| | | T=10 | T=20 | T=50 | T=10 | T=20 | T=50 |
| FFHQ 128 | LDM | 67.78 | 30.43 | 12.90 | | **4.87** | |
| | DDIM | 29.56 | 21.45 | 15.08 | 88.22 | 45.30 | 22.23 |
| | DiffAE | 20.80 | 16.70 | 12.57 | 12.59 | 9.23 | 5.93 |
| | JEDI | **18.41** | **14.38** | **12.26** | **11.50** | **8.17** | **5.48** |
| CelebA 64 | LDM | 41.87 | 31.40 | 25.80 | | **9.58** | |
| | DDIM | 16.38 | 12.70 | 8.52 | 78.44 | 20.20 | 16.76 |
| | DiffAE | 12.92 | 10.18 | 7.05 | 14.14 | 10.09 | 5.87 |
| | JEDI | **12.35** | **9.49** | **6.65** | **11.84** | **8.60** | **5.15** |

Table 1: Performance metrics for image generation (FID) and reconstruction (rFID). For reconstruction, LDM, utilizing only VAE, yields a single result for each metric, irrespective of $T$.

### 4.1.2 IMAGE INTERPOLATION

We perform interpolation within the latent space and then reconstruct images from the resulting interpolated representations. Given two latent vectors, $\mathbf{z}_0^1$ and $\mathbf{z}_0^2$, we employ linear interpolation using the formula: $\alpha \mathbf{z}_0^1 + (1 - \alpha)\mathbf{z}_0^2$, where $\alpha$ represents the interpolation ratio and we use $\alpha = 0.2, 0.4$. This interpolated latent vector is subsequently fed into the decoder to produce the interpolated image. We evaluate interpolation using 50k images from each dataset. Following this, we calculate the FID between the interpolated images and the original dataset, with results presented in Table 2. When considered alongside Table 1, several observations emerge: (1) LDM, while demonstrating commendable performance in reconstruction, falls short in maintaining quality during interpolation, as evidenced by its notably higher FID in interpolation; (2) Across all scenarios, JEDI consistently posts competitive performance, underscoring the compactness of our representation space. More details and case study are in §C.1.

### 4.1.3 IMAGE MANIPULATION

We deploy our model trained on FFHQ to CelebA-HQ in a zero-shot fashion. We select the CelebA-HQ dataset for this task due to the availability of 40 binary classification category labels. Following DiffAE, we train a linear classifier, $\mathbf{y} = \mathbf{w}^\top \mathbf{z}_0 + b$, on 70% of the training data for each attribute. This classifier predicts the attribute based on the representation $\mathbf{z}_0$. To construct the manipulated image representation, we use $\mathbf{z}_0' = \mathbf{z}_0 + \epsilon \mathbf{w}$,

| Dataset | Model | $\alpha = 0.2$ | | | $\alpha = 0.4$ | | |
|---|---|---|---|---|---|---|---|
| | | T=10 | T=20 | T=50 | T=10 | T=20 | T=50 |
| FFHQ 128 | LDM | | 21.29 | | | 75.13 | |
| | DDIM | 144.40 | 104.91 | 75.81 | 181.07 | 131.80 | 105.31 |
| | DiffAE | 13.25 | 11.33 | 9.38 | 22.93 | 23.01 | 22.17 |
| | JEDI | **12.57** | **9.80** | **6.66** | **19.11** | **18.36** | **16.98** |
| CelebA 64 | LDM | | 9.95 | | | 28.78 | |
| | DDIM | 148.22 | 80.06 | 51.84 | 163.26 | 103.01 | 82.77 |
| | DiffAE | 11.09 | 9.30 | 6.90 | 17.13 | 16.82 | 15.35 |
| | JEDI | **9.75** | **8.68** | **6.23** | **16.12** | **16.56** | **14.85** |

Table 2: Overall performance comparison of Image interpolation.

where $\epsilon$ is a scalar determining the manipulation magnitude. The manipulated representation $\mathbf{z}'$ is then used to generate the corresponding image. Our evaluation focuses on two aspects: 1) *Image Quality Post-Manipulation*: As shown in Table 3, JEDI consistently yields high-quality images after manipulation. 2) *Alignment with Target Class*: We test the linear classifier on the remaining 30% of the dataset. In terms of weighted AUC, JEDI's representation achieves 0.915, closely matching DiffAE's 0.917. Notably, JEDI surpasses in accuracy, registering 0.893 against DiffAE's 0.795. A detailed AUC comparison, enriched with case studies, can be found in §C.1.

## 4.2 TEXT

**Setup** We base our experiments on the LatentOps model (Liu et al., 2022a), a type of large VAE. We adopt BERT-small (Devlin et al., 2019; Bhargava et al., 2021) as the encoder and GPT2-xl (Radford et al., 2019) as the decoder. An preliminary warmup training is executed to align the encoder and decoder using the bookcorpus dataset (Zhu et al., 2015) in the absence of the diffusion model. Sub-

| Model | $\epsilon = 0.1$ | $\epsilon = 0.3$ |
|-------|---------|---------|
| DiffAE | 19.62 | 23.69 |
| JEDI | **13.48** | **18.43** |

Table 3: Manipulation.

sequently, joint training is undertaken on the Yelp review dataset (Shen et al., 2017; Li et al., 2018). To ensure a fair comparison, we primarily compare JEDI with VAE (Kingma & Welling, 2014; Li et al., 2020) and DAAE (Shen et al., 2020), maintaining consistent architecture and training procedures with the only difference in training objective. To holistically evaluate JEDI's performance across three core capabilities, in addition to generation and reconstruction assessments (§4.2.1), we have selected two downstream tasks: sentence *interpolation* (§4.2.2) and *style transfer* (§4.2.3). More details are in §C.2.1. For the evaluation, we use BLEU to measure content preservation, while MAUVE (Pillutla et al., 2021) assess fluency. In the context of text style transfer, we evaluate success rate using attribute accuracy from a BERT classifier.

### 4.2.1 TEXT GENERATION AND RECONSTRUCTION

First, we assess the foundational performance in both reconstruction and generation tasks. For reconstruction, we process the test set and evaluate its outcomes using the BLEU metric. For generation, we produce 100 sentences derived from the prior distribution and subsequently evaluate them using the MAUVE score. The results are presented in Table 4. JEDI not only enhances the reconstruction quality but also generates notably more fluent text. The main reason for our

| Method | Rec↑ | Gen↑ |
|--------|------|------|
| VAE | 87.6 | 0.20 |
| DAAE | 86.1 | 0.01 |
| JEDI | **92.1** | **0.98** |

Table 4: Reconstruction and generation results.

improvements is that the prior in VAE and DAAE can not properly reflect the real distribution of data in latent space. In our case, the real distribution is jointly defined by the diffusion model, which could better fit the diffusion.

### 4.2.2 SENTENCE INTERPOLATION

We randomly choose 200 samples from the test set. We then perform interpolation between the initial 100 samples and the remaining 100 samples. When provided with two samples, we encode them into $\mathbf{z}_0^1$ and $\mathbf{z}_0^2$. For decoding, given that both latent distributions adhere to a Gaussian distribution, we employ the spherical linear interpolation (Slerp) method (Shoemake, 1985) using the formula: $\text{Slerp}(\mathbf{z}_0^1, \mathbf{z}_0^2; t)$ with $0 \leq t \leq 1$. The expected outcome is a series of fluent sentences that exhibit a progressive semantic shift. The number of interpolation steps

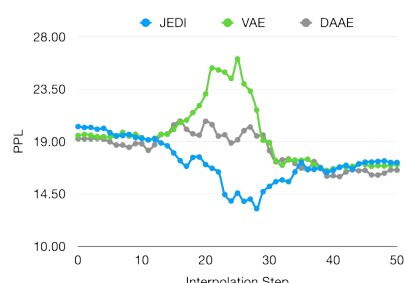

Figure 4: Perplexity of text interpolation.

is 50, and we evaluate the quality of each step to explore the performance. As illustrated in Fig. 4, JEDI consistently produces fluent sentences in the middle that align with the original data distribution. In contrast, baselines, especially VAE, exhibit poor generation at this midpoint. This can be attributed to the presence of holes in the VAE's latent space, leading to the generation of subpar samples. We also evaluate MAUVE and BLEU and provide the generated examples in §C.2.2.

### 4.2.3 STYLE TRANSFER VIA LATENT VECTOR ARITHMETIC

Previous research (Mikolov et al., 2013; Shen et al., 2020) demonstrate that representations of sentences or words, derived from unsupervised learning, can capture linguistic relationships through simple arithmetic operations, as exemplified by "King" - "Man" + "Woman" ≈ "Queen". In our study, we focus on the sentiment attribute to determine whether the representation of JEDI can draw a similar inference. We derive a single sentiment vector by separately averaging the latent vectors for 100 sentences each with positive and negative sentiments from the dataset. Subsequently, we compute the difference between the two sentiment vectors, denoted as $\mathbf{v}$. For a given sentence, it is first encoded to obtain its latent vector, $\mathbf{z} = \mathcal{E}(\mathbf{x})$.

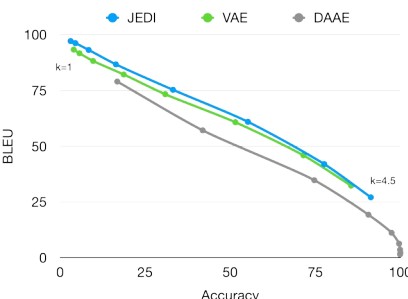

The sentiment-transferred sentence is then acquired via $\mathcal{D}(\mathbf{z} \pm k\mathbf{v})$, where $k$ serves as a weight modulating the degree of transfer. We tried different $k \in \text{range}(1, 5, .5)$ and summarize the results in Fig.5. With the same accuracy, JEDI achieves competitive BLEU, better than VAE and DAAE, which demonstrates the latent representation could capture linguistic relationships. While maintaining the same accuracy, JEDI attains a BLEU, outperforming both VAE and DAAE. This underscores that the latent representation of JEDI effectively captures linguistic relationships and semantics.

Figure 5: Results of style transfer.

## 4.3 PROTEIN

**Setup**  Following the setup of ReLSO (Castro et al., 2022), we adopt a simple transformer as the encoder and convolutional layers as the decoder. In line with ReLSO, we jointly train a simple regressor to predict the fitness values from the latent embeddings of protein sequences. This fitness value, representing the log of the round-to-round sequence frequency ratio, serves as a performance metric with higher values indicating superiority. Our models are trained and evaluated on the Gifford (Liu et al., 2019) dataset. We trained 2 separate models with $w$ in Eq. (16) set to 1 and 5, and also trained a vanilla VAE as one of our baselines. Detailed experimental setups are in §C.3.1. We evaluate JEDI's representation ability in §4.3.1 and its generation ability through protein optimization (§4.3.2). The evaluation on JEDI's reconstruction ability is provided in the appendix at §C.3.3.

### 4.3.1 PROTEIN REPRESENTATION

After training, each protein sequence in the test set is transformed into a latent representation, upon which the fitness value is predicted using the regressor in the latent space. Evaluation metrics including Mean Squared Error, L1 norm, Pearson correlation coefficient, and Spearman correlation coefficient are computed between the predicted values and the ground truth; these results are presented in Table 5. The regressor trained in the latent space of JEDI demonstrates superior performance over the regressor from the baseline models on both MSE and L1 norm. This suggests that our model obtained more refined representations for the protein sequences, leading to more accurate predictions by the regressor. This is also evident from the visualization of the latent space. In Fig. 6, we display the latent spaces for both ReLSO and JEDI, categorizing proteins by their respective fitness values intervals. In the latent space of ReLSO in Fig. 6a, proteins with fitness values less than 0 exhibits an overlap, and sequence with fitness value greater than 0.5 are interwined with each other. While some overlap persists across intervals in our latent space in Fig. 6b, the delineation between each interval is more clear.

### 4.3.2 PROTEIN OPTIMIZATION

We optimize a protein sequence by optimizing its corresponding representation in the latent space. We adopt the sampling algorithm introduced in LatentOps that solves an ODE involving the regressor (Liu et al., 2022b). This approach requires a target fitness value to guide the optimization. We set this value to 1, 1.5, 2, and 2.5, all of which represent reasonable fitness

| Model | MSE↓ | L1↓ | Pearson↑ | Spearman↑ |
|---|---|---|---|---|
| VAE | 0.253 | 0.365 | **0.839** | 0.474 |
| ReLSO | 0.293 | 0.401 | 0.826 | **0.477** |
| JEDI ($w$=1) | 0.248 | 0.364 | 0.821 | 0.465 |
| JEDI ($w$=5) | **0.234** | **0.355** | 0.833 | 0.460 |

Table 5: Protein Representation

values within the dataset. Visualizations of the optimized sequence using LatentOps optimization algorithms are shown in Fig. 7. The grey dots represent proteins from the training set, while colored dots represens optimized proteins with varying target fitness values. As shown in Fig. 7a, the pseudo-conves nature of ReLSO leads to convergence of optimized sequences at a singular point, revealing a diversity deficit. In contrast, as depicted in Fig. 7b, our model not only achieves superior fitness values but also fosters a broader protein variety. For a detailed quantitative analysis, refer to §C.3.2.

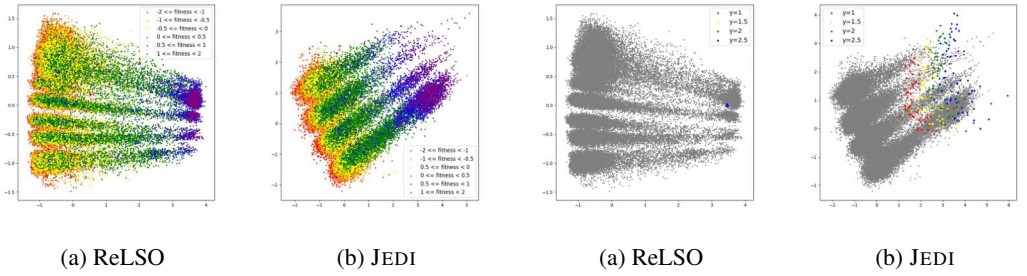

| (a) ReLSO | (b) JEDI | (a) ReLSO | (b) JEDI |

Figure 6: Protein Latent Space.                    Figure 7: Optimized protein sequence.

## 5 RELATED WORK

To address the limitations of existing generative models, the research community has focused on the development of hybrid and augmented architectures. One direction has been the fusion of VAEs (Razavi et al., 2019; Vahdat & Kautz, 2020) and GANs (Karras et al., 2020; Esser et al., 2021; Xia et al., 2023). This led to the emergence of VAE-GAN hybrids (Larsen et al., 2016; Xu et al., 2021), designed to enhance both generation and reconstruction capabilities. However, simultaneous achievement of all three fundamental abilities, namely, generation, reconstruction, and representation, in one unified model remains a challenging problem.

Recently, diffusion models (Sohl-Dickstein et al., 2015; Ho et al., 2020) have showcased impressive generative capabilities. However, they are limited by the absence of a robust semantic latent space, which curtails their versatility in broader applications. To bridge this gap, some researchers have explored the parameterization of the VAE's prior with diffusion models (Wehenkel & Louppe, 2021). For instance, Vahdat et al. (2021) leveraged a score-based model to parameterize the VAE's prior for image generation, introducing simplifications for tractability. Another noteworthy approach is the Latent Diffusion Model (LDM) (Rombach et al., 2021), or Stable Diffusion. This model integrates the architecture of an autoencoder with a diffusion model in the latent space, with the VAE's role being pivotal in image dimension compression, thereby optimizing diffusion training. A deeper dive into this is available in §D.

The importance of a semantically meaningful latent space has been underscored in various works (Donahue et al., 2017; Shen et al., 2020; Preechakul et al., 2022; Li et al., 2020; Liu et al., 2022b). An effective latent space not only enhances performance in reconstruction and generation but also accompanies semantic representations that are invaluable for downstream applications, including clustering, anomaly detection, among others.

## 6 CONCLUSION

In this work, we generalized the diffusion model to introduce JEDI, an innovative generative framework designed to seamlessly integrate the three core functionalities for generative models: generation, reconstruction and representation. To achieve the goad, we incorporated parameterized encoder and decoder transformations into the conventional diffusion process, redefining both the forward and reverse process. Subsequently, we derived an end-to-end training objective from the data likelihood, which encompasses three components, each addressing one of the core functionalities. Experimental results across image, text, and protein sequence data demonstrate that JEDI consistently outperforms strong baselines across all three core functionalities. Building on the versatility of our approach, which accommodates a range of data types via tailored encoder/decoder configurations, future investigations could extend its applicability across diverse domains. Additionally, a deeper exploration into the distinct benefits conferred by JEDI would be a valuable avenue of research.

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

# A DERIVATION

## A.1 DERIVATION OF OUR ELBO LOSS

Below is a derivation of Eq. (13).

$$\mathbb{E}_{\mathbf{x}\sim q(\mathbf{x})}\left[-\log p_{\phi,\theta}(\mathbf{x})\right] \tag{19}$$

$$\leq \mathbb{E}_{\mathbf{x}\sim q(\mathbf{x})}\left[-\log p_{\phi,\theta}(\mathbf{x}) + \mathrm{KL}(q_{\lambda}(\mathbf{z}_{0:T}|\mathbf{x})||p_{\phi,\theta}(\mathbf{z}_{0:T}|\mathbf{x}))\right] \tag{20}$$

$$= \mathbb{E}_{\mathbf{x}\sim q(\mathbf{x})}\left[-\log p_{\phi,\theta}(\mathbf{x}) + \mathbb{E}_{\mathbf{z}_{0:T}\sim q_{\lambda}(\mathbf{z}_{0:T}|\mathbf{x})}[\log \frac{q_{\lambda}(\mathbf{z}_{0:T}|\mathbf{x})}{p_{\phi,\theta}(\mathbf{z}_{0:T}|\mathbf{x})}]\right] \tag{21}$$

$$= \mathbb{E}_{\mathbf{x}\sim q(\mathbf{x})}\left[-\log p_{\phi,\theta}(\mathbf{x}) + \mathbb{E}_{\mathbf{z}_{0:T}\sim q_{\lambda}(\mathbf{z}_{0:T}|\mathbf{x})}[\log \frac{q_{\lambda}(\mathbf{z}_{0:T}|\mathbf{x})}{p_{\phi,\theta}(\mathbf{x}, \mathbf{z}_{0:T})/p_{\phi,\theta}(\mathbf{x})}]\right] \tag{22}$$

$$= \mathbb{E}_{\mathbf{x}\sim q(\mathbf{x})}\left[-\log p_{\phi,\theta}(\mathbf{x}) + \mathbb{E}_{\mathbf{z}_{0:T}\sim q_{\lambda}(\mathbf{z}_{0:T}|\mathbf{x})}[\log \frac{q_{\lambda}(\mathbf{z}_{0:T}|\mathbf{x})}{p_{\phi,\theta}(\mathbf{x}, \mathbf{z}_{0:T})}] + \log p_{\phi,\theta}(\mathbf{x})\right] \tag{23}$$

$$= \mathbb{E}_{\mathbf{x}\sim q(\mathbf{x}),\mathbf{z}_{0:T}\sim q_{\lambda}(\mathbf{z}_{0:T}|\mathbf{x})}\left[\log \frac{q_{\lambda}(\mathbf{z}_{0:T}|\mathbf{x})}{p_{\phi,\theta}(\mathbf{x}, \mathbf{z}_{0:T})}\right] \tag{24}$$

$$= \mathbb{E}_{q}\left[-\log \frac{p_{\phi,\theta}(\mathbf{x}, \mathbf{z}_{0:T})}{q_{\lambda}(\mathbf{z}_{0:T}|\mathbf{x})}\right] \tag{25}$$

## A.2 DERIVATION OF OUR KL LOSS

Below is a derivation of Eq. (14).

$$\mathcal{L}(\lambda, \phi, \theta) = \mathbb{E}_{q}\left[-\log \frac{p_{\phi,\theta}(\mathbf{x}, \mathbf{z}_{0:T})}{q_{\lambda}(\mathbf{z}_{0:T}|\mathbf{x})}\right] \tag{26}$$

$$= \mathbb{E}_{q}\left[-\log \frac{p_{\phi}(\mathbf{x}|\mathbf{z}_0)p_{\theta}(\mathbf{z}_{0:T})}{q(\mathbf{z}_{1:T}|\mathbf{z}_0)q_{\lambda}(\mathbf{z}_0|\mathbf{x})}\right] \tag{27}$$

$$= \mathbb{E}_{q}\left[-\log \frac{p_{\phi}(\mathbf{x}|\mathbf{z}_0)p(\mathbf{z}_T)\prod_{t=1}^{T}p_{\theta}(\mathbf{z}_{t-1}|\mathbf{z}_t)}{\prod_{t=1}^{T}q(\mathbf{z}_t|\mathbf{z}_{t-1})q_{\lambda}(\mathbf{z}_0|\mathbf{x})}\right] \tag{28}$$

$$= \mathbb{E}_{q}\left[-\log p(\mathbf{z}_T) - \sum_{t=2}^{T}\log \frac{p_{\theta}(\mathbf{z}_{t-1}|\mathbf{z}_t)}{q(\mathbf{z}_t|\mathbf{z}_{t-1})} - \log \frac{p_{\theta}(\mathbf{z}_0|\mathbf{z}_1)}{q(\mathbf{z}_1|\mathbf{z}_0)} - \log \frac{p_{\phi}(\mathbf{x}|\mathbf{z}_0)}{q_{\lambda}(\mathbf{z}_0|\mathbf{x})}\right] \tag{29}$$

$$= \mathbb{E}_{q}\left[-\log p(\mathbf{z}_T) - \sum_{t=2}^{T}\log \frac{p_{\theta}(\mathbf{z}_{t-1}|\mathbf{z}_t)}{q(\mathbf{z}_{t-1}|\mathbf{z}_t, \mathbf{z}_0)} \cdot \frac{q(\mathbf{z}_{t-1}|\mathbf{z}_0)}{q(\mathbf{z}_t|\mathbf{z}_0)} - \log \frac{p_{\theta}(\mathbf{z}_0|\mathbf{z}_1)}{q(\mathbf{z}_1|\mathbf{z}_0)} - \log \frac{p_{\phi}(\mathbf{x}|\mathbf{z}_0)}{q_{\lambda}(\mathbf{z}_0|\mathbf{x})}\right] \tag{30}$$

$$= \mathbb{E}_{q}\left[-\log \frac{p(\mathbf{z}_T)}{q(\mathbf{z}_T|\mathbf{z}_0)} - \sum_{t=2}^{T}\log \frac{p_{\theta}(\mathbf{z}_{t-1}|\mathbf{z}_t)}{q(\mathbf{z}_{t-1}|\mathbf{z}_t, \mathbf{z}_0)} - \log \frac{p_{\theta}(\mathbf{z}_0|\mathbf{z}_1)}{q_{\lambda}(\mathbf{z}_0|\mathbf{x})} - \log p_{\phi}(\mathbf{x}|\mathbf{z}_0)\right] \tag{31}$$

$$= \mathbb{E}_{q}\left[\underbrace{\mathrm{KL}\left(q(\mathbf{z}_T|\mathbf{z}_0)||p(\mathbf{z}_T)\right)}_{\mathcal{L}_T} + \sum_{t=2}^{T}\underbrace{\mathrm{KL}\left(q(\mathbf{z}_{t-1}|\mathbf{z}_t, \mathbf{z}_0)||p_{\theta}(\mathbf{z}_{t-1}|\mathbf{z}_t)\right)}_{\mathcal{L}_{t-1}} \right.$$
$$\left. + \underbrace{\mathrm{KL}\left(q_{\lambda}(\mathbf{z}_0|\mathbf{x})||p_{\theta}(\mathbf{z}_0|\mathbf{z}_1)\right)}_{\mathcal{L}_{\mathrm{align}}} - \underbrace{\log p_{\phi}(\mathbf{x}|\mathbf{z}_0)}_{\mathcal{L}_{\mathrm{rec}}}\right]. \tag{32}$$

## A.3 DERIVATION OF EACH KL TERM

In line with Ho et al. (2020), our objective exclusively involves KL divergences between Gaussians, thus facilitating closed-form evaluations.

**For $\mathcal{L}_T$** : The posterior distribution $q$ lacks learnable parameters due to the deterministic forward mapping from $\mathbf{z}_0$ to $\mathbf{z}_T$. Specifically, we have $q(\mathbf{z}_T|\mathbf{z}_0) = \mathcal{N}(\mathbf{z}_T; \sqrt{\bar{\alpha}_T}\mathbf{z}_0, (1-\bar{\alpha}_T)\mathbf{I})$. When $\bar{\alpha}_T \approx 1$, this simplifies to $q(\mathbf{z}_T|\mathbf{z}_0) = \mathcal{N}(\mathbf{z}_T; \mathbf{0}, \mathbf{I})$. Given this property, $\mathcal{L}_T$ remains constant during training and can be excluded from optimization.

**For** $\mathcal{L}_{1:T-1}$ : This term aligns with the conventional diffusion model in Eq. (7). Given the two conditional Gaussian distributions, $p_\theta(\mathbf{z}_{t-1}|\mathbf{z}_t)$ in Eq. (12) and $q(\mathbf{z}_{t-1}|\mathbf{z}_t, \mathbf{z}_0)$ derived as

$$q(\mathbf{z}_{t-1}|\mathbf{z}_t, \mathbf{z}_0) = \mathcal{N}(\mathbf{z}_{t-1}; \tilde{\boldsymbol{\mu}}(\mathbf{z}_t, \mathbf{z}_0), \tilde{\beta}_t), \tag{33}$$

$$\text{where} \quad \tilde{\boldsymbol{\mu}}_t(\mathbf{z}_t, \mathbf{z}_0) := \frac{\sqrt{\bar{\alpha}_{t-1}}\beta_t}{1-\bar{\alpha}_t}\mathbf{z}_0 + \frac{\sqrt{\alpha_t}(1-\bar{\alpha}_{t-1})}{1-\bar{\alpha}_t}\mathbf{z}_t \quad \text{and} \quad \tilde{\beta}_t := \frac{1-\bar{\alpha}_{t-1}}{1-\bar{\alpha}_t}\beta_t, \tag{34}$$

$\mathcal{L}_{t-1}$ can be compactly represented as:

$$\mathcal{L}_{t-1} = \mathbb{E}_q\left[\frac{1}{2\sigma_t^2}\|\tilde{\boldsymbol{\mu}}_t(\mathbf{z}_t, \mathbf{z}_0) - \boldsymbol{\mu}_\theta(\mathbf{z}_t, t)\|^2\right] + C, \tag{35}$$

where $C$ is a constant, independent with $\theta$. We can expand Eq. (35) further by reparameterizing $q(\mathbf{z}_t|\mathbf{z}_0)$ as $\mathbf{z}_t(\mathbf{z}_0, \boldsymbol{\epsilon}) = \sqrt{\bar{\alpha}_t}\mathbf{z}_0 + \sqrt{1-\bar{\alpha}_t}\boldsymbol{\epsilon}$ for $\boldsymbol{\epsilon} \sim \mathcal{N}(\mathbf{0}, \mathbf{I})$ and reparemetrizing $\boldsymbol{\mu}_\theta(\mathbf{z}_t, t)$ as $\frac{1}{\sqrt{\bar{\alpha}_t}}(\mathbf{z}_t - \sqrt{1-\bar{\alpha}_t}\boldsymbol{\epsilon}_\theta(\mathbf{z}_t, t))$:

$$\mathcal{L}_{t-1} = \mathbb{E}_{\mathbf{z}_0 \sim q(\mathbf{z}_0), \boldsymbol{\epsilon} \sim \mathcal{N}(\mathbf{0}, \mathbf{I})}\left[\gamma_t\left\|\boldsymbol{\epsilon} - \boldsymbol{\epsilon}_\theta\left(\sqrt{\bar{\alpha}_t}\mathbf{z}_0 + \sqrt{1-\bar{\alpha}_t}\boldsymbol{\epsilon}, t\right)\right\|^2\right], \tag{36}$$

where $\gamma_t = \frac{\beta_t^2}{2\sigma_t^2\alpha_t(1-\bar{\alpha}_t)}$, which aligns with the $\gamma_t$ in Eq. (7).

**For** $\mathcal{L}_{\text{align}}$ : This loss serves to align the latent representations derived from both the encoder and the diffusion model, ensuring consistent latent spaces across the framework. From Eq. 10 and Eq. 12, we know that $p_\theta(\mathbf{z}_0|\mathbf{z}_1)$ and $q_\lambda(\mathbf{z}_0|\mathbf{x})$ are Gaussian distributions. Therefore, the KL divergence has the similar form as Eq. (35), while the $\tilde{\boldsymbol{\mu}}_t(\mathbf{z}_t, \mathbf{z}_0)$ becomes the mean of $q_\lambda(\mathbf{z}_0|\mathbf{x})$, which corresponds to $\mathcal{E}_\lambda(\mathbf{x})$. So the loss function becomes:

$$\mathcal{L}_{\text{align}} = \mathbb{E}_q\left[\frac{1}{2\sigma_1^2}\|\mathcal{E}_\lambda(\mathbf{x}) - \boldsymbol{\mu}_\theta(\mathbf{z}_1, 1)\|^2\right] \tag{37}$$

$$= \mathbb{E}_q\left[\frac{\beta_1^2}{2\sigma_1^2\alpha_t(1-\bar{\alpha}_1)}\frac{\alpha_1(1-\bar{\alpha}_1)}{\beta_1^2}\|\mathcal{E}_\lambda(\mathbf{x}) - \boldsymbol{\mu}_\theta(\mathbf{z}_1, 1)\|^2\right] \tag{38}$$

$$= \mathbb{E}_q\left[\gamma_1 \cdot \rho\|\mathcal{E}_\lambda(\mathbf{x}) - \boldsymbol{\mu}_\theta(\mathbf{z}_1, 1)\|^2\right] \tag{39}$$

where $\gamma_1$ is consistent with Eq. (36), $\rho := \frac{\alpha_1(1-\bar{\alpha}_1)}{\beta_1^2}$ is a constant, and $\boldsymbol{\mu}_\theta(\mathbf{z}_1, 1)$ is reparemeterized under the same reparemetrization trick used in $\mathcal{L}_{1:T-1}$, i.e., $\boldsymbol{\mu}_\theta(\mathbf{z}_1, 1) = \frac{1}{\sqrt{\bar{\alpha}_1}}(\mathbf{z}_1 - \sqrt{1-\bar{\alpha}_1}\boldsymbol{\epsilon}_\theta(\mathbf{z}_1, 1))$.

**For** $L_{\text{rec}}$ : This is the reconstruction loss corresponding to the first term in Eq. (8). Depending on the data type, it can be formulated using different loss functions. In our model, we employ MSE loss for continuous data like images and cross-entropy for discrete data, including texts and proteins.

### A.4 DERIVATION OF OUR FINAL LOSS

The final objective in Eq. (14) in can be reformulated as:

$$\mathbb{E}_{q,\boldsymbol{\epsilon}}\left[\sum_{t=2}^{T}\gamma_t\|\boldsymbol{\epsilon} - \boldsymbol{\epsilon}_\theta(\mathbf{z}_t, t)\|^2 + \gamma_1 \cdot \rho\|\mathcal{E}_\lambda(\mathbf{x}) - \boldsymbol{\mu}_\theta(\mathbf{z}_1, 1)\|^2 + \mathcal{L}_{\text{rec}}\right] \tag{40}$$

$$= \mathbb{E}_{q,\boldsymbol{\epsilon}}\left[\sum_{t=2}^{T}\gamma_t\|\boldsymbol{\epsilon} - \boldsymbol{\epsilon}_\theta(\mathbf{z}_t, t)\|^2 + \gamma_1 \cdot \rho\|\mathcal{E}_\lambda(\mathbf{x}) - \boldsymbol{\mu}_\theta(\mathbf{z}_1, 1)\|^2 + T \cdot \frac{1}{T}\mathcal{L}_{\text{rec}}\right] \tag{41}$$

$$= \mathbb{E}_{q,\boldsymbol{\epsilon}}\left[\sum_{t=2}^{T}\gamma_t\left(\|\boldsymbol{\epsilon} - \boldsymbol{\epsilon}_\theta(\mathbf{z}_t, t)\|^2 + \frac{1}{\gamma_t T}\mathcal{L}_{\text{rec}}\right) + \gamma_1\left(\rho\|\mathcal{E}_\lambda(\mathbf{x}) - \boldsymbol{\mu}_\theta(\mathbf{z}_1, 1)\|^2 + \frac{1}{\gamma_1 T}\mathcal{L}_{\text{rec}}\right)\right] \tag{42}$$

We further introduce a hyperparameter $w$ to balance the diffusion/align loss with the reconstruction loss, which gives us the following loss:

$$\mathbb{E}_{q,\boldsymbol{\epsilon}}\left[\sum_{t=2}^{T}\gamma_t\left(w\|\boldsymbol{\epsilon} - \boldsymbol{\epsilon}_\theta(\mathbf{z}_t, t)\|^2 + \frac{1}{\gamma_t T}\mathcal{L}_{\text{rec}}\right) + \gamma_1\left(w\left(\rho\|\mathcal{E}_\lambda(\mathbf{x}) - \boldsymbol{\mu}_\theta(\mathbf{z}_1, 1)\|^2\right) + \frac{1}{\gamma_1 T}\mathcal{L}_{\text{rec}}\right)\right]. \tag{43}$$

## A.5    DERIVATION OF OUR LOSS FROM A LEARNABLE PRIOR PERSPECTIVE

The following is a derivation of Eq. (18) in §3.4, the objective from a learnable prior perspective.

$$\mathcal{L}(\lambda, \phi, \theta; \mathbf{x}) = \mathbb{E}_q \left[ -\log \frac{p_{\phi,\theta}(\mathbf{x}, \mathbf{z}_{0:T})}{q_\lambda(\mathbf{z}_{0:T}|\mathbf{x})} \right] \tag{44}$$

$$= \mathbb{E}_q \left[ -\log \frac{p_\phi(\mathbf{x}|\mathbf{z}_0)p_\theta(\mathbf{z}_{0:T})}{q_\lambda(\mathbf{z}_0|\mathbf{x})q(\mathbf{z}_{1:T}|\mathbf{z}_0)} \right] \tag{45}$$

$$= \mathbb{E}_q \left[ -\log \frac{p_\phi(\mathbf{x}|\mathbf{z}_0)p_\theta(\mathbf{z}_0)}{q_\lambda(\mathbf{z}_0|\mathbf{x})} \right] \tag{46}$$

$$= -\mathbb{E}_{q_\lambda(\mathbf{z}_0|\mathbf{x})}[\log p_\phi(\mathbf{x}|\mathbf{z}_0)] + \mathrm{KL}(q_\lambda(\mathbf{z}_0|\mathbf{x})||p_\theta(\mathbf{z}_0)) \tag{47}$$

## B    ALGORITHM

Below shows the complete training algorithm of JEDI.

---
**Algorithm 1** Training

---
1: **repeat**
2:     $\mathbf{x} \sim q(\mathbf{x})$
3:     $\boldsymbol{\epsilon}_0 \sim \mathcal{N}(0, \mathbf{I})$
4:     $\mathbf{z}_0 = \mathcal{E}_\lambda(\mathbf{x}) + \beta_0 \boldsymbol{\epsilon}_0$
5:     $t \sim \mathrm{Uniform}(\{1, \ldots, T\})$
6:     $\boldsymbol{\epsilon} \sim \mathcal{N}(0, \mathbf{I})$
7:     $\mathbf{z}_t = \sqrt{\bar{\alpha}_t}\mathbf{z}_0 + \sqrt{1 - \bar{\alpha}_t}\boldsymbol{\epsilon}$
8:     **if** t == 1 **then**
9:         $\boldsymbol{\mu}_\theta(\mathbf{z}_1, 1) = \frac{1}{\sqrt{\bar{\alpha}_1}}(\mathbf{z}_1 - \sqrt{1 - \bar{\alpha}_1}\boldsymbol{\epsilon}_\theta(\mathbf{z}_1, 1))$
10:        Take gradient descent step on $\mathcal{L}_1^{\mathrm{final}}$
11:    **else**
12:        Take gradient descent step on $\mathcal{L}_t^{\mathrm{final}}$
13:    **end if**
14: **until** converged

---

## C    DETAILS OF EXPERIMENTS

This section is the counterpart of the experiment section §4.

### C.1    IMAGE

#### C.1.1    SETUP

**Model Architecture**    In line with the architecture presented by Diffusion Autoencoders (DiffAE) (Preechakul et al., 2022), our model is structured with an encoder designed as a UNet and a decoder functioning as a conditional diffusion-based model at the pixel level. Given the latent semantic representation $\mathbf{z}_0$ and a random Gaussian sample $\mathbf{x}_T$ that shares the same dimensionality as the raw data $\mathbf{x}$, the decoder employs reverse diffusion transitions to produce the output $\hat{\mathbf{x}}$. Complementing this, we integrate an additional standard diffusion process, with transitions realized through a straightforward MLP fortified with skip connections.

**Dataset**    Following the approach of DiffAE, we train our model and subsequently evaluate its reconstruction and generation capabilities on the FFHQ (Karras et al., 2019) and CelebA (Karras et al., 2018) datasets. To assess the representation ability, we employ the CelebA-HQ dataset (Karras et al., 2018). The FFHQ dataset comprises 70,001 images, CelebA encompasses 202,599 images, and CelebA-HQ consists of 9,000 images distributed across 40 categories.

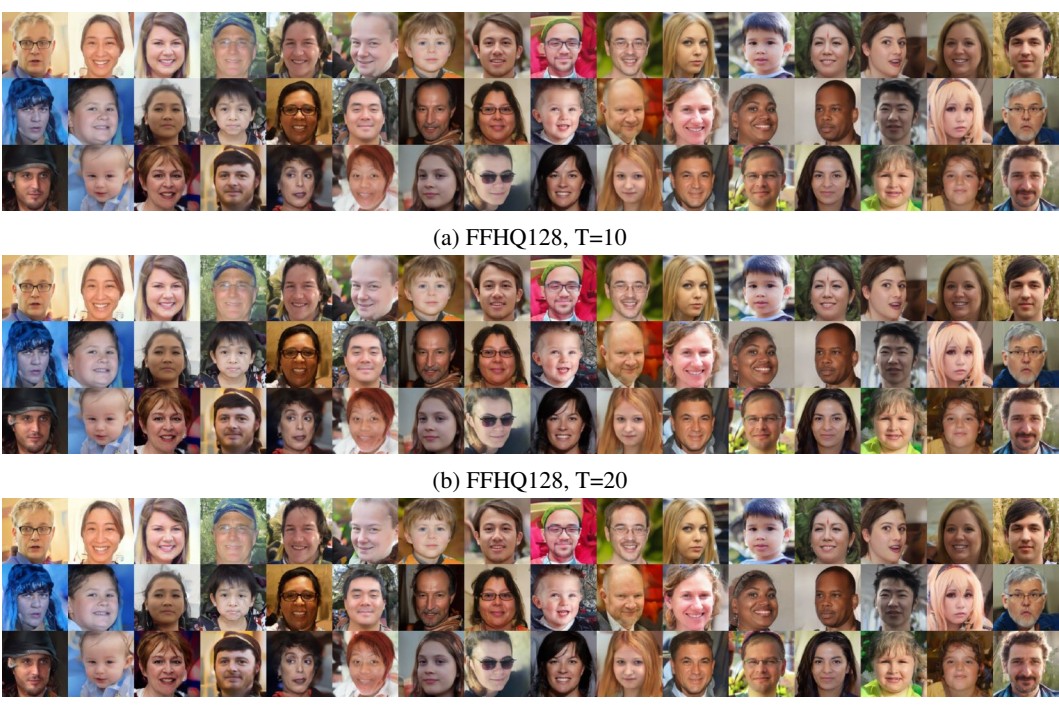

(a) FFHQ128, T=10

(b) FFHQ128, T=20

(c) FFHQ128, T=50

**Hyperparameter settings**   We trained our model on two Nvidia A100-SXM4-40GB GPUs with a batch size of 100. For evaluation purposes, we sampled 50,000 images to compute the FID, setting total steps $T = 100$ for both the diffusion process and the decoder at every 500,000 training steps interval. The optimization was carried out using the Adam Optimizer, with a learning rate of $1 \times 10^{-4}$ and no weight decay. The image dimensions inputted into the model were consistently set at $128 \times 128$ for FFHQ and $64 \times 64$ for CelebA.

**Evaluation Metrics**   For the assessment of the generated images' quality, we resort to the Fréchet Inception Distance (FID) (Heusel et al., 2017), a widely-accepted metric in the field. To assess the fidelity of our reconstruction, we employ the reconstruction-FID (rFID) metric.

### C.1.2   IMAGE GENERATION

For FFHQ128, we present the generated images in Figure 8a, 8b, 8c). Then for CelebA64, the generated images are shown in Figure 9a, 9b, 9c. As depicted in the figures, images generated with $T = 50$ typically exhibit finer granularity when contrasted against those produced with $T = 10$ or $T = 20$.

### C.1.3   IMAGE RECONSTRUCTION

For a comparative analysis, we present reconstructed images from various models, each utilizing a distinct total step $T$ in the decoder. The results for FFHQ128 and CelebA64 are depicted in Figure 10 and Figure 11, respectively. From the figures, it's clear that the VAE in LDM is effective at reconstruction. Our model JEDI also produces strong results.

### C.1.4   IMAGE REPRESENTATION

**Interpolation**   The interpolation results for FFHQ128 and CelebA64 are illustrated in Figure 12 and Figure 13, respectively. A close examination of the figures reveals that while the VAE in LDM is adept at reconstruction, its interpolation with $\alpha = 0.4$ appears akin to a superposition of two images. This aligns with the inherent nature of their VAE, where the representation predominantly encodes spatial rather than semantic information. In contrast, both our approach and DiffAE yield

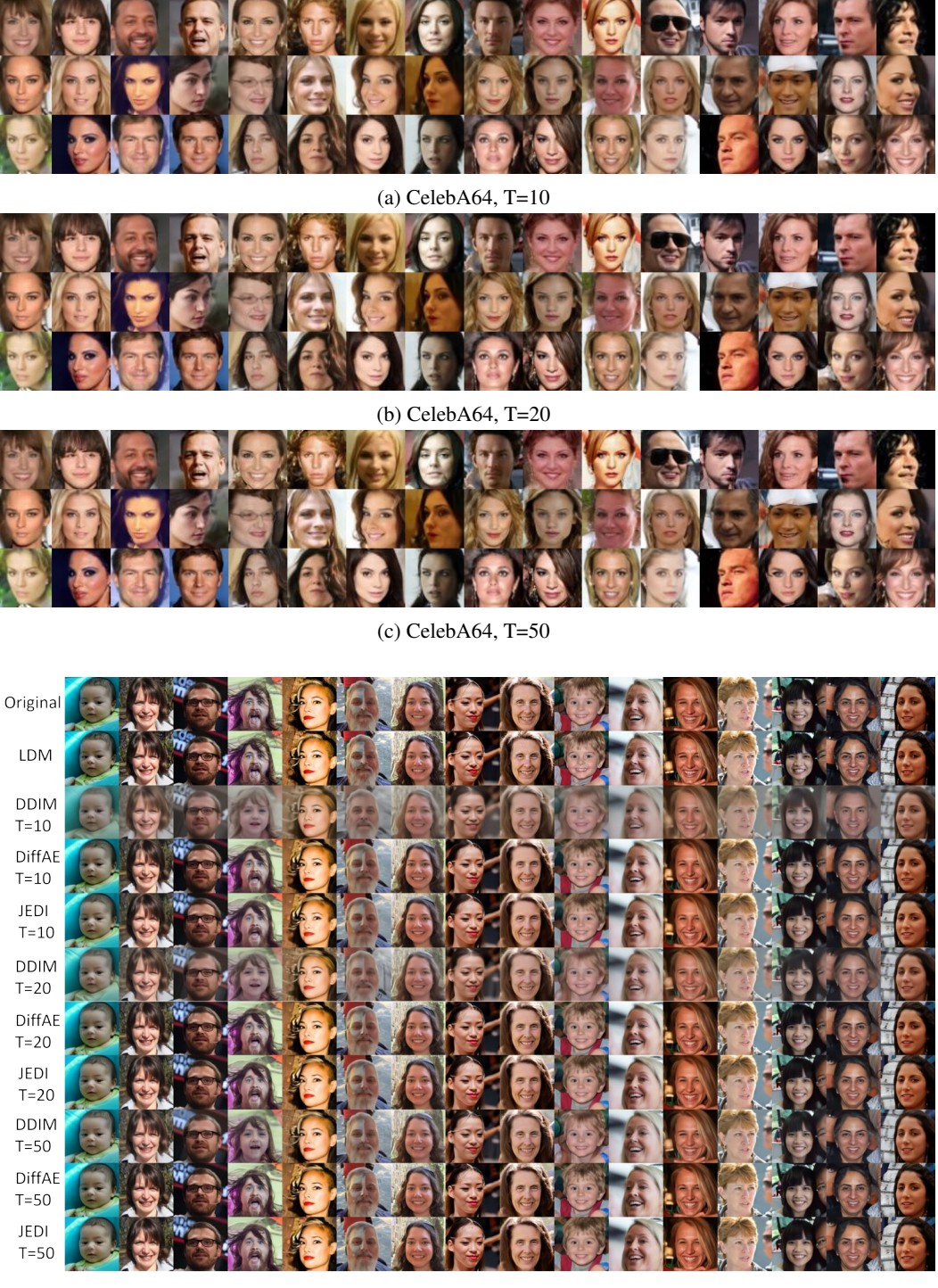

(a) CelebA64, T=10

(b) CelebA64, T=20

(c) CelebA64, T=50

Figure 10: Image reconstructions for FFHQ128 with different models. Best viewed with zooming in.

superior results at $\alpha = 0.4$. Specifically, our method demonstrates fewer visual artifacts compared with DiffAE, underscoring the better representation space of our model.

**Manipulation   1. AUC Comparison:** The comprehensive comparisons are presented in Table 6. As indicated by the table, our model consistently delivers comparable AUC values across all classes.

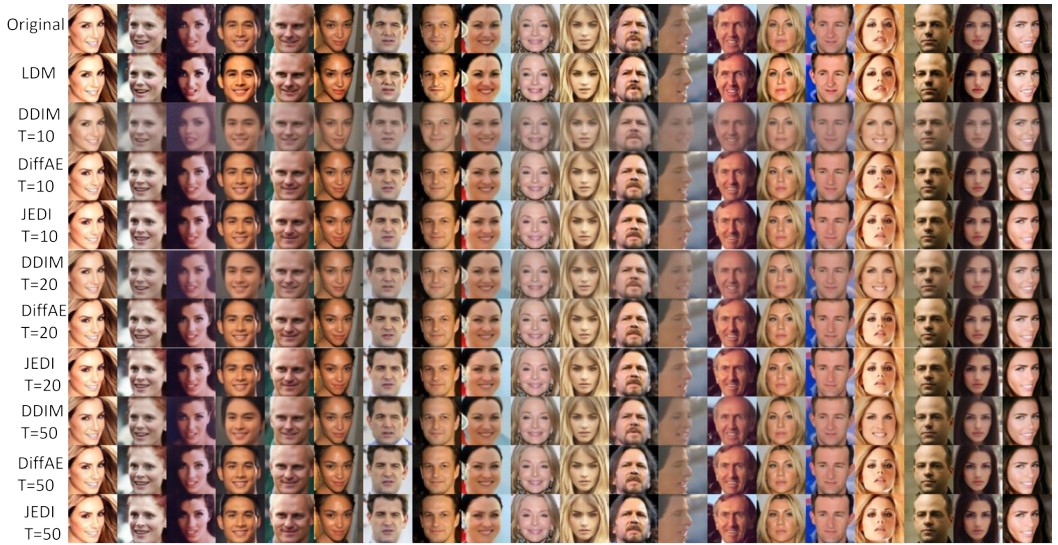

Figure 11: Image reconstructions for CelebA64 with different models. Best viewed with zooming in.

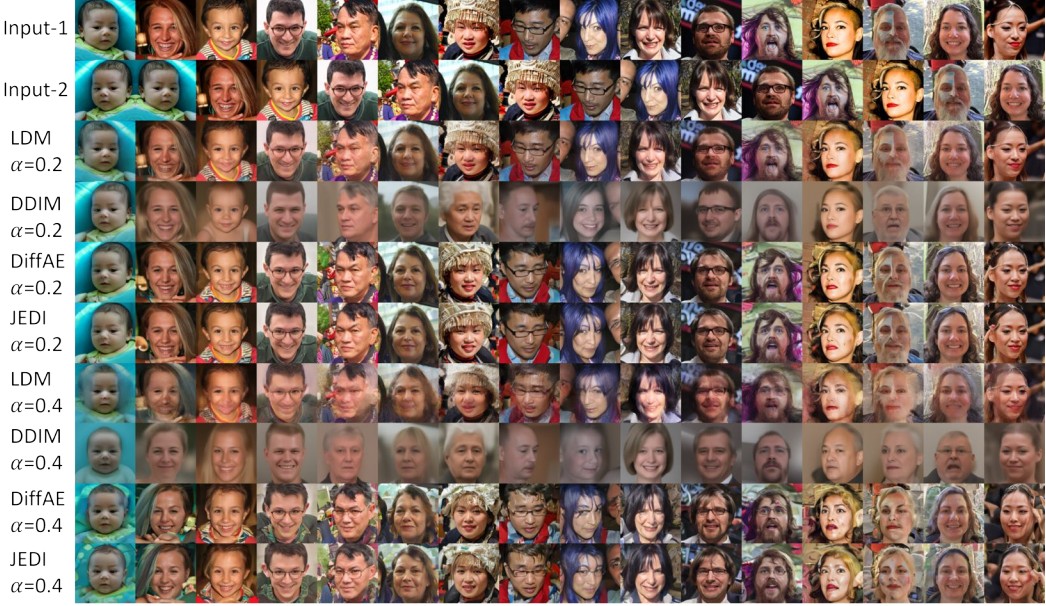

Figure 12: Image interpolations for FFHQ128 with different models. $T$ is fixed to 50 in these experiments.

**2. Case Study:** Manipulated images derived from JEDI and the baseline models are showed in Figure 14. The presented examples suggest that our approach better handles semantic information, resulting in more proficient manipulations.

## C.2 TEXT

### C.2.1 SETUP

**Model Architecture** We closely adhere to the experimental setup presented in LatentOps (Liu et al., 2022a). For our encoder, denoted as $\mathcal{E}_\lambda$, we utilize the BERT-small model (Devlin et al.,

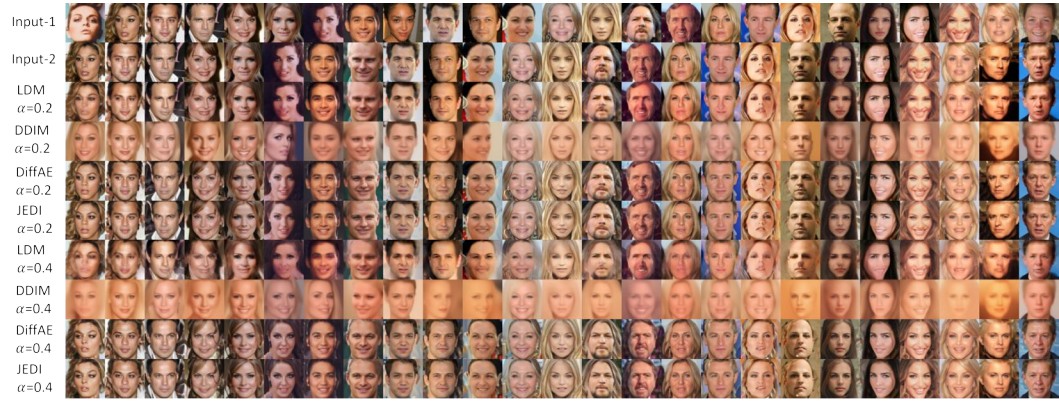

Figure 13: Image interpolations for CelebA64 with different models. $T$ is fixed to 50 in these experiments.

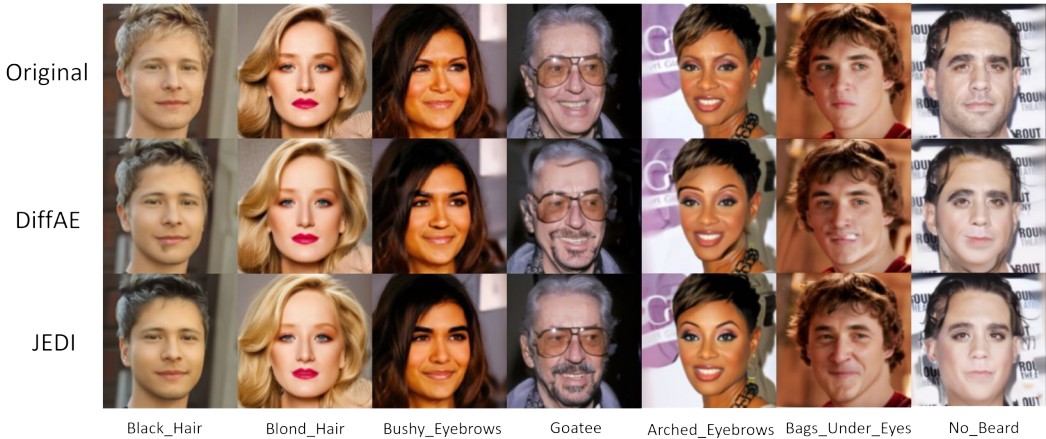

Figure 14: **Image Manipulation:** The procedure for manipulation is detailed in Section 4.1.3. The class names provided at the bottom indicate the target class towards which the images are being manipulated.

2019; Bhargava et al., 2021). As for the decoder, represented as $\mathcal{D}_\phi$, we employ the GPT2-xl architecture (Radford et al., 2019). Our diffusion model is constructed using a straightforward MLP with skip connections, as inspired by (Preechakul et al., 2022). The latent dimension is set to 128.

Building upon the methodologies of Li et al. (2020); Liu et al. (2022a), we equip the pretrained language model (LM) with a linear layer that precedes the LM, facilitating the passage of $z_0$ to the decoder. To maintain generative capabilities and acclimate the LM to the latent space, we incorporate an additional transformer layer between the original first layer and the embedding layer of the LM, fostering adaptability. During the training phase, our optimization is confined to the MLP layers, the embedding layer, the newly inserted transformer layer, the encoder, and the diffusion model with all other parameters remaining frozen.

**Dataset** Regarding our dataset selection, we commence with the bookcorpus dataset (Zhu et al., 2015) to train the autoencoder in the absence of the diffusion model. Subsequently, we engage in joint training of the model with diffusion, utilizing the Yelp review dataset (Shen et al., 2017), which has been preprocessed by Li et al. (2018). It's noteworthy that Yelp serves as a sentiment dataset, encompassing approximately 179K negative and 268K positive sentences.

**Baselines** To ensure a fair comparison, we primarily contrast our approach with VAE (Kingma & Welling, 2014; Li et al., 2020) and DAAE (Shen et al., 2020), maintaining consistent architecture

| Class | # Positives | DiffAE | JEDI |
|---|---|---|---|
| 5_o_Clock_Shadow | 1314 | 0.9469 | 0.9466 |
| Arched_Eyebrows | 3171 | 0.8822 | 0.8811 |
| Attractive | 5009 | 0.8849 | 0.8792 |
| Bags_Under_Eyes | 2528 | 0.8787 | 0.8821 |
| Bald | 174 | 0.9886 | 0.9843 |
| Bangs | 1593 | 0.9779 | 0.9770 |
| Big_Lips | 3187 | 0.7031 | 0.7108 |
| Big_Nose | 2796 | 0.8757 | 0.8683 |
| Black_Hair | 1895 | 0.9483 | 0.9427 |
| Blond_Hair | 1469 | 0.9775 | 0.9760 |
| Blurry | 27 | 0.8434 | 0.8762 |
| Brown_Hair | 2074 | 0.8476 | 0.8334 |
| Bushy_Eyebrows | 1658 | 0.9174 | 0.9117 |
| Chubby | 579 | 0.9439 | 0.9339 |
| Double_Chin | 481 | 0.9503 | 0.9486 |
| Eyeglasses | 404 | 0.9916 | 0.9877 |
| Goatee | 650 | 0.9724 | 0.9690 |
| Gray_Hair | 346 | 0.9865 | 0.9856 |
| Heavy_Makeup | 3963 | 0.9714 | 0.9695 |
| High_Cheekbones | 4043 | 0.9490 | 0.9474 |
| Male | 3184 | 0.9977 | 0.9969 |
| Mouth_Slightly_Open | 4125 | 0.9784 | 0.9777 |
| Mustache | 495 | 0.9573 | 0.9547 |
| Narrow_Eyes | 1061 | 0.8569 | 0.8600 |
| No_Beard | 7047 | 0.9784 | 0.9754 |
| Oval_Face | 1772 | 0.7494 | 0.7469 |
| Pale_Skin | 458 | 0.9635 | 0.9618 |
| Pointy_Nose | 2738 | 0.7263 | 0.7235 |
| Receding_Hairline | 718 | 0.9373 | 0.9340 |
| Rosy_Cheeks | 975 | 0.9482 | 0.9416 |
| Sideburns | 684 | 0.9768 | 0.9726 |
| Smiling | 4050 | 0.9827 | 0.9803 |
| Straight_Hair | 1866 | 0.8030 | 0.8027 |
| Wavy_Hair | 3099 | 0.8804 | 0.8770 |
| Wearing_Earrings | 2328 | 0.8933 | 0.8851 |
| Wearing_Hat | 311 | 0.9875 | 0.9862 |
| Wearing_Lipstick | 4911 | 0.9803 | 0.9789 |
| Wearing_Necklace | 1496 | 0.7788 | 0.7789 |
| Wearing_Necktie | 600 | 0.9583 | 0.9560 |
| Young | 6871 | 0.9230 | 0.9130 |
| Weighted Avg AUC | - | **0.9174** | 0.9154 |
| Weighted Avg ACC | - | 0.7954 | **0.8929** |

Table 6: Classification performance comparison.

and training procedures. The sole distinction lies in the training objective. DAAE (Denoising Adversarial Autoencoders) is a modified autoencoder that improves text generation by reconstructing sentences from slightly altered versions, combining adversarial training with a denoising objective.

**Tasks**  In addition to assessing individual functionalities (reconstruction and generation), our goal is to empirically evaluate the holistic performance of JEDI across the three foundational capabilities. To this end, we primarily focus on two downstream tasks: text style transfer and interpolation. The reasons are shown below:

- Good Reconstruction: Central to this is the ability to preserve content. In the context of text style transfer and interpolation, maintaining the integrity of the original content is paramount.

- Robust Representation: Beyond mere content preservation, it's crucial to accurately capture the intrinsic meaning, nuances, and essential features of the input text. This capability becomes especially significant in tasks that require a deep understanding of the source material, such as text style transfer.

- Potent Generation: The ultimate measure of a model's efficacy is its output. For JEDI, ensuring the fluency and coherence of the generated text is a testament to its robust generation capabilities, which is evident in both the text style transfer and interpolation tasks.

Thus, these two tasks can synthetically reflect the three functionalities.

### C.2.2 SENTENCE INTERPOLATION

In addition to perplexity, we assess using MAUVE and BLEU metrics, as depicted in Fig.15. For a more intuitive comparison, we present the generated sentences from intermediate steps in Table 7 and Table 7. It's evident that the transition in JEDI is smoother, and the intermediate sentences are more coherent.

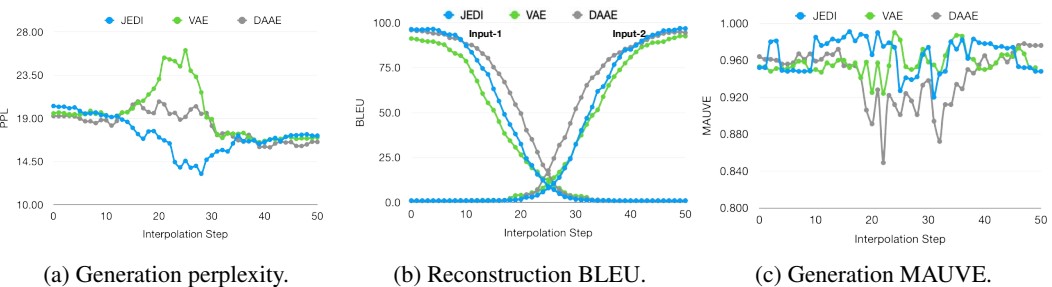

(a) Generation perplexity.  (b) Reconstruction BLEU.  (c) Generation MAUVE.

Figure 15: Results of sentence interpolation.

| Input-1 | i ca n't believe how inconsiderate this pharmacy is . |
| Input-2 | we were sat right away and every staff member was extremely friendly and happy . |

| | |
|---|---|
| VAE | i ca n't believe how inconsiderate this pharmacy is .
i was n't so worried about how pharmacy membership is an inconvenience .
i was n't worried about how inconsiderate this pharmacy is .
i was n't asked how inconsiderate this pharmacy is .
i was n't asked how trustworthy this pharmacy is .
i was shocked how inconsiderate this pharmacy is .
i was shocked ... how inconsiderate this pharmacy is .
i was shocked ... how inconsiderate this staff is extremely friendly .
i was sent ... how inconsiderately friendly and extremely disappointed .
we were sat right away and every staff member was extremely friendly and accommodating .
we were sat right away and every staff member was extremely friendly and welcoming .
we were sat right away and every staff member was extremely friendly and happy . |
| DAAE | i ca n't believe how inconsiderate this pharmacy is .
i ca n't believe how very inconsiderate this pharmacy is .
i ca n't believe how very inconsiderate this store is .
i could n't believe how very inconsiderate this store is .
i could n't believe how very inconvenient this restaurant is .
i could n't believe how very indifferent this pharmacy staff is .
i could not say how very inconvenient this store was friendly .
i could not say how very inconvenient the staff is being and i am very disappointed .
i could not say how very inconvenient the staff was being and i am very disappointed .
i could not say how very inconvenient the staff was being so friendly .
i could not say how very inconvenient the staff was extremely friendly and welcoming .
i could not say very much about the locker owner and was very welcomed .
i could not sit down and every friend was charging it was very uncomfortable .
we were so rudely referred to every good gas company and was extremely rude .
we were so rudely sat at every friend and it was extremely friendly .
we were sat down and every friend and staff was extremely happy .
we were sat down very much and the staff member was extremely friendly and welcoming .
we were sat right away and every staff member was extremely happy .
we were sat right away and every staff member was extremely friendly and welcoming .
we were sat right away and every staff member was extremely friendly and happy . |
| JEDI | i ca n't believe how inconsiderate this pharmacy is .
i ca n't believe how inconsiderate this staff is .
i 'm not sure how this pharmacy is .
i 'm not impressed how inconsiderate this place is .
i am very impressed and every customer service is great .
i was very impressed and every customer service was great .
i was very impressed and every staff member was amazing .
we were very impressed and every staff member was extremely friendly .
we were very impressed and every staff member was extremely friendly and helpful .
we were very impressed ... every staff member was extremely friendly and helpful .
we were sat right away and every staff member was extremely friendly and helpful .
we were sat right away and every staff member was extremely friendly and happy . |

Table 7: Examples of sentence interpolation with 50 steps. To avoid redundancy, repetitive outputs from adjacent steps are omitted.

| Input-1 | she was not happy being there . |
|---------|---------------------------------|
| Input-2 | the chips and guacamole were excellent too ! |

| VAE | she was not happy being there .
the chips were cheap too .
the chips were cheap and definitely excellent .
the chips were cheap and excellent !
the chips and guacamole were excellent !
the chips and guacamole were excellent too ! |
|------|---|

| DAAE | she was not happy being there .
she was not happy with the food being there .
she was not happy with the food and was not there .
she was so happy and had fun being there .
she was so happy and the food was excellent too .
they were so happy with the food and it was delicious !
the food was sooooo delicious and i was happy to have found it .
the food was sooooo delicious and i was happy to have found it !
the food was sooooo delicious and i was very happy !
the chips and guacamole were excellent too ! |
|-------|---|

| JEDI | she was not happy being there .
she was not happy .
she was not happy !
she was happy !
the food was excellent !
the chips and salsa were excellent too !
the chips and guacamole were excellent too ! |
|-------|---|

Table 8: Examples of sentence interpolation with 50 steps. To avoid redundancy, repetitive outputs from adjacent steps are omitted.

## C.3 PROTEIN

Protein design plays a crucial role in drug discovery, protein therapeutics and various other applications in biotechnology. However, due to the complex and large search space of protein sequences, traditional empirical methods that demands intensive and thorough experiments and screening for validation are expensive and time-consuming. Recent advances in machine learning and computational methods introduces new approaches for protein optimization, however, most of these methods focuses on protein design in the discrete text space and lacks a meaningful latent space. Our work combines the Autoencoder and Diffusion Model, enabling effective protein generation/optimization and establishing a robust latent space for protein representation.

### C.3.1 SETUP

**Architecture** We adopt the set up of ReLSO Castro et al. (2022) which consists of a simple transformer as the encoder and convolutional layers as the decoder. While ReLSO relies on Negative Sampling Loss which augments the datasets with synthetic samples with negative labels and Interpolative Regularization which penalizes differences between interpolated points and nearest neighbors to achieve a smooth pseudo-convex latent space, we leverage the same MLP with skip connections in our text and image model as the backbone of the diffusion model to act as a learnable prior. In line with ReLSO, we jointly train a simple regressor consists of linear layers and ReLU that predicts the fitness value with the latent embedding of a protein sequence. The regressor is trained with mean squared error and added to each sample loss of Eq. (16) :

$$\mathcal{L}_t^{\text{protein}} = \mathcal{L}_t^{\text{final}} + \mathcal{L}^{\text{regressor}}, \tag{48}$$

The dimension of the latent space is set to 30.

**Dataset** The models are trained on the Gifford Liu et al. (2019) dataset which was generated from directed evolution of $10^{10}$ mutants of an antibody against a single target(Ranibizumab) through three rounds of phage display panning. Fitness value is defined as the log of the round-to-round ratio of sequence frequencies and a higher fitness value indicate better performance. In this dataset, each protein sequence has a length of 20 with a vocabulary of 20 amino acids which are represented by 20 letters. The resulting dataset consists of 57603 sequences in the training set, 10166 sequences in the validation set, and 22690 sequences in the test set.

**Baselines** We have trained 2 separate models with $w$ in Eq. (16) set to 1 and 5. We have also trained a vanilla Variational Autoencoder as a baseline model. All the models are trained with the training set and evaluated with the test set.

### C.3.2 PROTEIN OPTIMIZATION

We optimize a protein sequence by optimizing its corresponding embedding in the latent space. Given a protein sequence $\mathbf{x}$, we first obtain its latent embedding with $\mathbf{z}_0 = \mathcal{E}_\lambda(\mathbf{x})$. $\mathbf{z}_0$ can be optimized with 100 steps of gradients of the regressor (GA: Gradient Ascent). Alternatively, we adopt the sampling algorithm introduced in LatentOps that solves an ODE involving the regressor Liu et al. (2022b). This approach requires a target fitness value to guide the optimization. We set this value to 1, 1.5, 2, and 2.5, all of which represents reasonable fitness value within the dataset. For evaluation, we optimize 60 random protein sequence and evaluated their fitness value using the regressor. We assess the results on Diversity, quantified by the average Levenshtein distance of each sequence relative to the other 59 optimized sequences; on Novelty, determined by the median of the minimum Levenshtein distance between each optimized sequence and the training set; and on Quality, measured by the negative log likelihood given by ProtGPT2, a large protein language model trained on the UniRef50 dataset.

The results are presented in Table 9. From Table 9, our model attains higher fitness values than ReLSO when using the same settings. Note that since the latent space of our model is non-convex, optimizing a sequence using gradient ascent without stopping criteria could result in proteins with unnaturally high fitness values.

| Model | Algorithm | Max Fitness | Mean Fitness | Diversity | Novelty | NLL |
|---|---|---|---|---|---|---|
| VAE | GA | 1.821 | -0.316 | 12.955 | 1 | 29.841 |
| VAE | LatentOps | 1.000 | 0.824 | 13.242 | 5.5 | 29.747 |
| ReLSO | GA | 0.739 | 0.579 | 12.795 | 7 | 29.022 |
| ReLSO | LatentOps | 0.738 | 0.737 | 2.121 | 6 | 28.631 |
| JEDI ($w$=1) | GA | 4.012 | 2.283 | 12.801 | 7 | 29.803 |
| JEDI ($w$=1) | LatentOps | 1.002 | 1.000 | 13.294 | 6 | 29.825 |
| JEDI ($w$=5) | GA | 9.769 | 7.674 | 12.816 | 10 | 28.938 |
| JEDI ($w$=5) | LatentOps | 1.003 | 1.000 | 13.072 | 5 | 29.433 |
| ReLSO (target fitness=1.5) | LatentOps | 0.7382 | 0.7371 | 2.376 | 7 | 28.748 |
| ReLSO (target fitness=2.0) | LatentOps | 0.7386 | 0.7373 | 2.362 | 7 | 28.803 |
| ReLSO (target fitness=2.5) | LatentOps | 0.7383 | 0.7372 | 2.027 | 7 | 28.823 |
| JEDI ($w$=1, target fitness=1.5) | LatentOps | 1.502 | 1.500 | 12.907 | 6 | 28.713 |
| JEDI ($w$=1, target fitness=2.0) | LatentOps | 2.002 | 1.999 | 12.580 | 6.5 | 29.705 |
| JEDI ($w$=1, target fitness=2.5) | LatentOps | 2.505 | 2.500 | 12.466 | 7 | 29.775 |

Table 9: Comparison of Protein Optimization.

| Model | Cross Entropy | Reconstructed Bleu |
|---|---|---|
| VAE | 1.011 | 0.651 |
| ReLSO | 0.940 | 0.734 |
| JEDI (w=1) | 0.996 | 0.515 |
| JEDI (w=5) | 1.113 | 0.660 |

Table 10: Protein Reconstruction

### C.3.3 PROTEIN RECONSTRUCTION

Reconstruction is evaluated with Cross Entropy and BLEU to compare input and reconstructed protein sequences in the test set. The results are presented in Table 10. From Table 10, JEDI achieves comparable reconstruction ability compared to the baselines.

## D CONTRASTING WITH LATENT DIFFUSION MODELS

Latent Diffusion Models (LDMs), often termed Stable Diffusion Rombach et al. (2021), utilize an architectural combination of an autoencoder and a diffusion model in latent space. In this section, we delineate the primary distinctions between LDMs and our proposed approach in detail:

**Autoencoder's Functionality:** *LDMs:* The overarching objective of their autoencoder is twofold: to compress images into a compact latent representation and to ensure robust reconstruction capabilities from these latent vectors. To bias the autoencoder towards stronger reconstruction, they introduce a KL term but assign it a minute KL weight ($\sim 10^{-6}$). Their employment of a purely convolution-based encoder-decoder emphasizes spatial preservation in the latent space, which, while bolstering reconstruction, poses challenges to distilling semantically rich latent features. *Our Approach:* Our autoencoder extends beyond mere dimensionality reduction. It's intricately tailored to synchronize effectively with the diffusion process, thereby fostering a more semantically-coherent latent space. Instead of relying on the conventional KL regularization against a standard Gaussian, we harness a learnable prior, rendering our latent space more adaptive and insightful.

**Training Paradigm:** *LDMs:* Their training strategy bifurcates into two discrete phases: initial autoencoder training followed by subsequent training of the diffusion model in latent space. Consequently, the latent space's architecture predominantly adheres to the objectives set forth by the autoencoder. *Our Approach:* Our methodology pivots on end-to-end training, ensuring the latent space's architecture is sculpted by the holistic objectives of the entire model. This integrated approach instills the latent space with nuanced semantics and more discernible significance, enhancing both interpretability and utility.

