# OpenReview forum: "Generation, Reconstruction, Representation All-in-One: A Joint Autoencoding Diffusion Model"
_ICLR.cc/2024/Conference — ICLR 2024 Conference Withdrawn Submission_

### Official Review · Reviewer_oXqa · 2023-10-29

**Soundness:** 4 excellent
**Presentation:** 3 good
**Contribution:** 3 good
**Rating:** 6
**Confidence:** 3

**Summary:**

The paper presents Joint Autoencoding Diffusion (JEDI), a novel generative framework that unifies the three core capabilities of deep generative models: generation, reconstruction, and representation. JEDI introduces parameterized encoder/decoder transformations between raw data and compact representations, which are learned jointly with all other diffusion model parameters. The model demonstrates strong performance in diverse tasks and data modalities, including images, text, and proteins.

**Strengths:**

1. JEDI provides a unified framework that integrates the three core capabilities of deep generative models, offering versatile applications.

2. The model demonstrates strong performance across diverse tasks and data modalities, outperforming existing specialized deep generative models.

3. JEDI can naturally accommodate discrete data, such as text and protein sequences, which have been difficult for diffusion models.

**Weaknesses:**

1. The paper could provide more insights into the theoretical foundations and connections between JEDI and other generative models.

2. The experimental results could be expanded to include more data modalities and tasks to further demonstrate JEDI's versatility.

3. The paper could provide a more in-depth discussion on the differences between JEDI and other diffusion models, such as LDM, and the reasons for the observed performance differences.

**Questions:**

1. How does JEDI compare to other hybrid models, such as VAE-GAN hybrids, in terms of performance and versatility?

2. Are there any potential limitations or challenges in applying JEDI to other domains or tasks not covered in the paper?

3. Can the authors provide a more detailed analysis of the differences between JEDI and LDM, and the reasons for the observed performance differences?

4. According to my intuition, using low-dimensional latent variables and Euclidean distance for reconstruction may lead to blurry image generation. How does JEDI avoid this issue, and what is the specific dimensionality of the latent variables used?

---

### Official Review · Reviewer_Y7KF · 2023-10-31

**Soundness:** 2 fair
**Presentation:** 3 good
**Contribution:** 1 poor
**Rating:** 3
**Confidence:** 4

**Summary:**

This paper extends the diffusion models with the goal of achieving accurate reconstruction and rich latent representation beyond the generation capabilities of diffusion models. To do that, the model uses ideas from VAEs (which consists another family of generative models). The concrete idea behind the model is to learn the diffusion process along with the embedding process. The proposed model, called JEDI, is then evaluated on image generation, manipulation and reconstruction, along with text reconstruction and generation and protein design.

**Strengths:**

+ The figures in this work are informative, e.g. Fig. 1.

+ The paper is understandable, which helps the audience to grasp the ideas.

+ Experiments beyond the image modality, where diffusion models are popular, are conducted.

**Weaknesses:**

There are several weaknesses of this work, starting from the fact that it’s not clear what the purpose of this work is. Sure, you can propose a model that unifies different objectives, but what is the point of this model? Is it stronger than previous ones in generation? Or in reconstruction?

In addition, the paper mentions that it introduces a “unified generative model”. Does this mean that the proposed model can be used as a GAN? Or express a GAN or an auto-regressive model?

**Questions:**

The paper uses the MSE loss as used in VAEs in the past. However, won’t this MSE loss result in blurry images?

I find the empirical validation of this paper to be rather weak. Firstly, the compared works are some of the fundamental papers in diffusion models and not the state-of-the-art in the domain. Aren’t there any more recent papers on the domain?

Similarly, the state-of-the-art beyond the diffusion models are not evaluated here, while the papers mentions that it is a unified generative model. For instance, how about comparing with StyleGAN?

What is the computational trade-off in time and in parameters in the tables? There is no mention of parameters or time comparisons.

I find that CelebA64 is selected, which is a low-resolution variant, while diffusion models (and especially latent diffusion) were designed for high-resolution images, so currently the experiments do not seem a fair comparison for the baselines.

In addition, why isn’t there any comparison with ImageNet or other standard datasets of diffusion models?

In text modality, one of the standards the last few years is auto-regressive models, but those are not included in the comparisons, could the authors elaborate on that?

Lastly, I have the impression that conditional GANs/VAEs can also achieve all of the stated goals, right? Could the authors elaborate on why those are not included in the comparisons?

---

### Official Review · Reviewer_cqsF · 2023-11-01

**Soundness:** 3 good
**Presentation:** 3 good
**Contribution:** 1 poor
**Rating:** 3
**Confidence:** 3

**Summary:**

The authors propose a diffusion framework that allows for simultaneous generation, reconstruction, and latent representation capabilities. This is achieved via a latent diffusion model where the encoder / decoder are trained simultaneously with the diffusion model.

**Strengths:**

- Simple formulation. The proposed model, while simple, is well described and easy to understand with clear figures and motivations.

- Multi-modal empirical evaluation. The authors evaluate the diffusion model on a range of tasks over several different modalities.

**Weaknesses:**

- Limited novelty. It is not clear to me how the proposed model (JEDI) really differs from prior latent diffusion approaches such as [1] and [2], outside of the inclusion of joint VAE / diffusion model training --- which is not that large of a conceptual leap. Perhaps the most important question is whether joint training makes that large of a difference. I think, at least in comparison to [1] and [2], this can be best understood by comparing against these two works in more standard benchmarks.

- Nonstandard benchmarks. The benchmarks in Section 4, especially 4.1.1, are not standard. Being face datasets, CelebA and FFHQ are generally considered easier benchmarks than, say, CIFAR10 and ImageNet, and T=10, 20, and 50 are somewhat low in terms of the number of timesteps for the diffusion models listed. Moreover, what is the latent dimension of the JEDI model in Table 1?

- Limited baselines. Why is JEDI not compared to more recent diffusion models (e.g. EDM, VDM) in the image domain, autoregressive / transformer models in the text domain, or more recent protein based models (e.g. SMCDiff, EDM).

At present, while the research direction is interesting, I do not believe that the novelty and empirical evaluation clear the bar for acceptance.

[1] NVAE: A Deep Hierarchical Variational Autoencoder. https://arxiv.org/abs/2007.03898

[2] High-Resolution Image Synthesis with Latent Diffusion Models. https://arxiv.org/abs/2112.10752

**Questions:**

Between Eqs 3 and 4: Should $q(x_t|x_{t−1})$ be $q(x_{t-1}|x_t)$? I believe that authors are referring to the reverse diffusion Markov chain; the forward diffusion is always well-defined, with or without $x_0$.

In Section 3.3, the reconstruction algorithm for JEDI is described as $D_\phi(\Epsilon_\lambda))$. However, in Table 1, there are separate reconstruction scores for different time horizons $T$. Why is this the case?

What is the latent dimensionality of the models used in reconstruction in Sections 4.1.1 and 4.2.1, and how do they compare to the latent dimensionalities of the competing models?

How is $w$ chosen in the diffusion loss?